# A Triangular Grid Filter Method Based on the Slope Filter

**Chuanli Kang** [1,2]**, Zitao Lin** [1,*] **, Siyi Wu** [1]**, Yiling Lan** [1]**, Chongming Geng** [1] **and Sai Zhang** [1]

1   College of Geomatics and Geoinformation, Guilin University of Technology, Guilin 541004, China;
    2014012@glut.edu.cn (C.K.)
2   Key Laboratory of Spatial Information and Geomatics, Guilin University of Technology, Guilin 541004, China
*   Correspondence: 2120211878@glut.edu.cn

**Abstract:** High-precision ground point cloud data has a wide range of applications in various fields, and the separation of ground points from non-ground points is a crucial preprocessing step. Therefore, designing an efficient, accurate, and stable ground extraction algorithm is highly significant for improving the processing efficiency and analysis accuracy of point cloud data. The study area in this article was a park in Guilin, Guangxi, China. The point cloud was obtained by utilizing the UAV platform. In order to improve the stability and accuracy of the filter algorithm, this article proposed a triangular grid filter based on the Slope Filter, found violation points by the spatial position relationship within each point in the triangulation network, improved KD-Tree-Based Euclidean Clustering, and applied it to the non-ground point extraction. This method is accurate, stable, and achieves the separation of ground points from non-ground points. Firstly, the Slope Filter is used to remove some non-ground points and reduce the error of taking ground points as non-ground points. Secondly, a triangular grid based on the triangular relationship between each point is established, and the violation triangle is determined through the grid; thus, the corresponding violation points are found in the violation triangle. Thirdly, according to the three-point collinear method to extract the regular points, these points are used to extract the regular landmarks by the KD-Tree-Based Euclidean Clustering and Convex Hull Algorithm. Finally, the dispersed points and irregular landmarks are removed by the Clustering Algorithm. In order to confirm the superiority of this algorithm, this article compared the filter effects of various algorithms on the study area and filtered the 15 data samples provided by ISPRS, obtaining an average error of 3.46%. The results show that the algorithm presented in this article has high processing efficiency and accuracy, which can significantly improve the processing efficiency of point cloud data in practical applications.

**Keywords:** slope filter; point cloud; triangular grid; KD-tree-based euclidean clustering

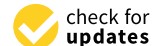



## 1. Introduction

Airborne Lidar technology combines a global positioning system, a laser scanner, and an inertial navigation system, which can obtain high-precision ground point cloud data quickly and efficiently [1]. Airborne Lidar point cloud data are an important data source for obtaining spatial three-dimensional information. Their ability to accurately express the location of ground objects means that they are widely used in many fields, such as constructing a digital elevation model (DEM), contour production, and three-dimensional model generation. However, due to the massive and disordered characteristics of point cloud data, the classification and feature extraction of point cloud data are particularly important; therefore, a point cloud filter is undoubtedly an important process.

Various studies have compared the current mainstream filter algorithms. Sithole used eight filter algorithms to filter different point cloud data and compared their errors, concluding that the adaptive irregular triangulation filter method has the best filter effect [2]. Zou compared and analyzed three commonly used filter methods from both qualitative and quantitative aspects and obtained the most suitable point cloud filter algorithm for

different terrain data [3]. These studies demonstrate that the point cloud filter algorithm has received extensive attention from scholars.

At present, the commonly used point cloud filter algorithms mainly include Segmentation-Based Filtering, the Cloth Simulation Filter, and the Slope Filter algorithm [4]. The progressive triangulation filter algorithm is proposed by Lin; the principle is to select the lowest point in the local area as the seed point of TIN and then use the remaining points to iteratively encrypt the triangular mesh [5]. However, this method cannot usually preserve the points on the scarp when meeting steep slopes, misjudging the non-ground points close to the ground as ground points. The Cloth Simulation Filter, proposed by Zhang, flips the original point cloud data up and down and then projects the flipped point cloud data and the simulated cloth node to the same horizontal plane. Following this, it finds the point corresponding to each node in the cloth and records the height of the corresponding point before projection; finally, comparing the distance between the laser point cloud and the cloth node in the same horizontal coordinate, if the distance exceeds the set threshold, it marks it as a non-ground point; otherwise, it is marked as a ground point [6]. The Cloth Simulation Filter has five adjustable parameters to complete the point cloud filter process. In the actual filter process, the filter effect is mainly improved by modifying the two parameters of grid resolution and cloth hardness, but the Cloth Simulation Filter has a poor filter effect with multi-story buildings. The Slope Filter is proposed by Vosselman the principle is to distinguish ground points from non-ground points according to the difference in slope [7]. However, the method of a fixed slope is not accurate enough to be applied to the whole situation. Improvements to the three commonly used filter methods have also been proposed. Sithole improves the Slope Filter so that the slope threshold can change with the terrain, which enhances the applicability of the algorithm [2]. However, this method requires multiple parameters to be set manually. In order to improve the accuracy, efficiency, and adaptability of the point cloud filter, Zhu also proposed an adaptive threshold point cloud filter method based on multi-level moving surface fitting [8]. This algorithm can obtain a good filter effect in complex and continuous terrain areas, but the filter effect in discontinuous terrain areas needs to be improved. Wang also proposed a Multi-scale Adaptive Point Cloud Slope Filter method based on the Slope Filter. This method referenced the idea of the multi-scale virtual network and used distance weighting to realize a multi-scale adaptive point cloud slope filter [9]. However, this method is less stable than a triangulation filter in urban areas.

The present study determined the non-ground points by finding the violation points and using these points for KD-Tree-Based Euclidean Clustering. The non-ground features are expressed in the form of cluster points. Various scholars have investigated the extraction of objects by clustering methods, and region growing [10] and K-means algorithms [11] have been employed for direct clustering in the 3-D model [12]. Rodriguez and Laio proposed clustering by rapid search and identification of density peaks [13]. Chen applied the exponential function density clustering method to indoor object extraction and achieved good results [14]. These studies showed that it is feasible to use a clustering algorithm for object extraction. Sun used hierarchical Euclidean Clustering to extract the building rooftop patches [15], Guo used KD-Tree-Based Euclidean Clustering for point cloud extraction and segmentation [12], and Gamal used DGCNN and Euclidean Clustering to finish the segmentation of Lidar buildings [16]. These studies prove that the Euclidean Clustering method is feasible for the extraction of regular buildings, and KD-Tree-Based Euclidean Clustering can increase the clustering operation speed. This method can be applied to the segmentation of large scene point clouds; consequently, this study proposed an extraction method for regular landmarks by using KD-Tree-Based Euclidean Clustering, calculating the violation point and collinear judgment to reduce the number of seeds that need indexing, and reducing the calculation amount.

In summary, the current ground-point filtering methods have certain limitations. Most of the filtering methods still use the grid method; the filtering effect of the grid method is easily affected by the size of the grid, and it is difficult to achieve better results for the complex building point cloud scene. The method of adaptively adjusting the size of the

grid is also more complex and inefficient. Therefore, it is necessary to propose a filtering method that is not bound by the size of the grid. The present study proposed a triangular mesh filter method based on the Slope Filter (Figure 1). This method finds the violation points based on the distance and angle relationship in the triangular grid; subsequently, it uses KD-Tree-Based Euclidean Clustering to find the non-ground points block through the spatial distance relationship.

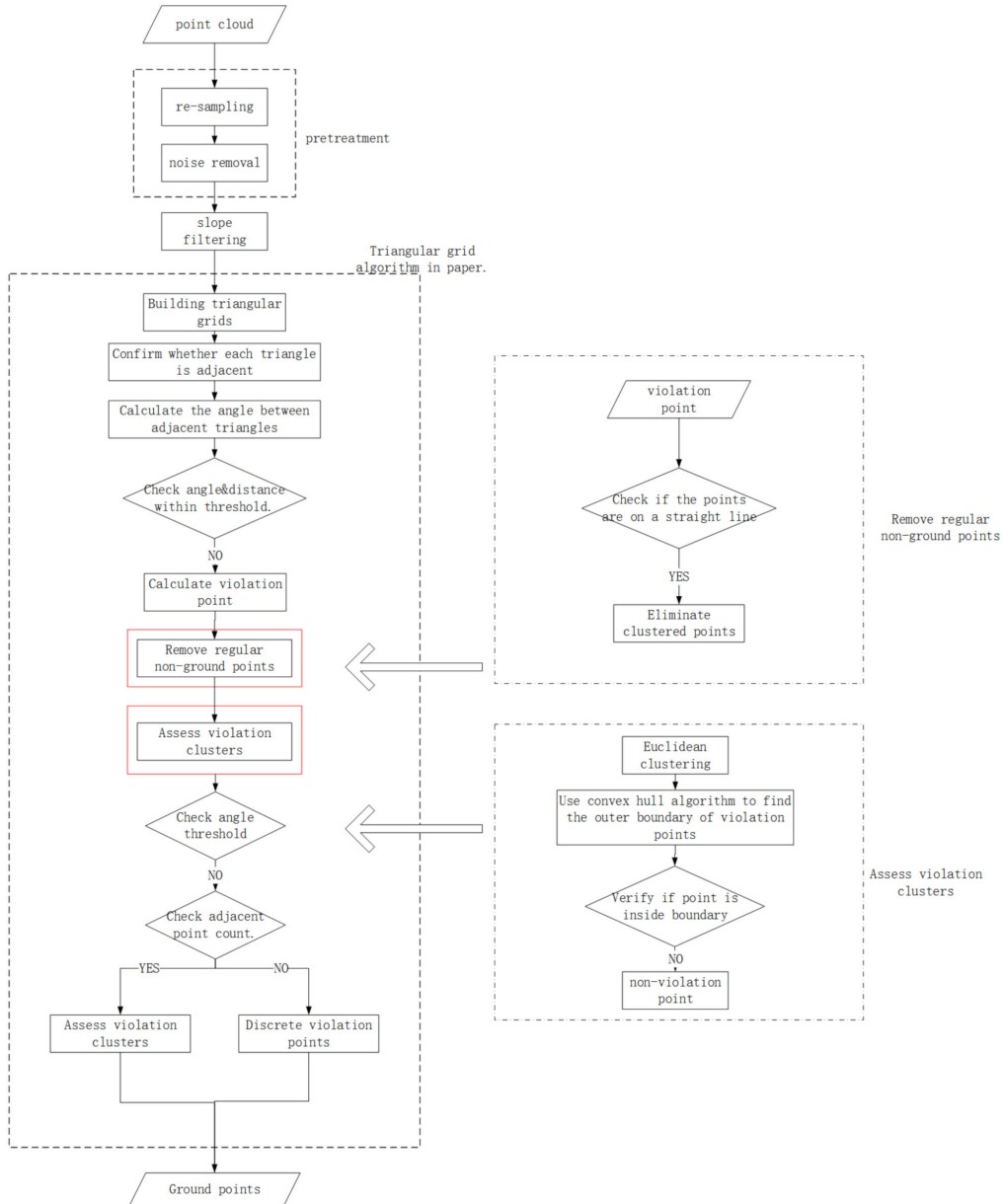

**Figure 1.** The algorithm flowchart.

## 2. The Algorithm Principle

### 2.1. Data Preprocessing

Due to factors such as the number of echoes set and occlusion during point cloud acquisition, the point cloud density may be too high or too low. Therefore, before processing point cloud data, resampling is required. Resampling can be up-sampling or down-sampling—up-sampling involves interpolating signals, while down-sampling involves extracting signals. The present experiment mainly explores the point cloud filter; therefore, only down-sampling of the research data was needed, and we used the setting distance for this process with the point cloud data. Specifically, we divide

the point cloud data into a three-dimensional grid and keep all points in each grid. Through down-sampling, the point cloud density can be reduced, the calculation during subsequent filter processes can be reduced, and the calculation speed can be improved. During the working process of Lidar, abnormal noise points may be generated due to various factors such as wind, water vapor, and illumination. This study used the Statistical Filter method to remove the noise points. The principle of this method is to calculate the average distance, μ, from each point to its neighboring points and determine the distance threshold $d_{threshold}$ based on the normal distribution. If μ is less than $d_{threshold}$, the point is considered an interior point and kept. If μ is more than $d_{threshold}$, the point is considered a noise point and removed.

### 2.2. The Slope Filter

Slope is a variable representing the degree of surface undulation; the meaning of the Slope Filter is to judge whether the grid slope value in the set grid is within the set range, and on this basis, to determine whether the grid is a ground point. This method was first proposed by Vosselman and improved by Sithole to make it suitable for steep terrain. The principle of the Slope Filter method is to establish a two-dimensional grid for the whole test area; each grid records the lowest point elevation of the internal storage point cloud to obtain the elevation grid (Figure 2), obtain elevation values within a two-dimensional horizontal grid in sequence (Figure 3), and subsequently calculate the slope value within the grid (*SlopeVaule*, Equation (1)), judging the attributes of the grid sequentially (Figure 4), and extracting the point cloud with the ground attribute. If the height difference between the point cloud and the lowest point in the grid is within the threshold, the point cloud can be considered the ground points.

$$SlopeValue = (z - \min_p(z))/sqrt((x - \min_p(x))^2 + (y - \min_p(y))^2) \qquad (1)$$

Here, $\min_p(x)$, $\min_p(y)$, $\min_p(z)$ are the grid positions with a lower elevation and their corresponding elevation values in the elevation grid; $x, y, z$ are the grid positions with a higher elevation and their corresponding elevation values in the elevation grid.

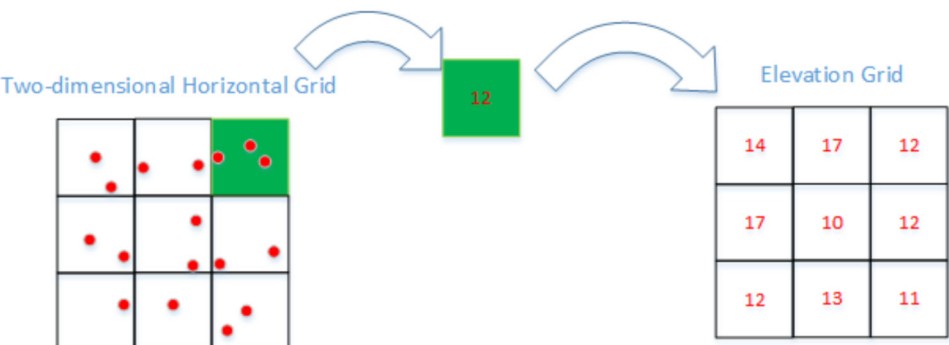

**Figure 2.** How to obtain an elevation grid.

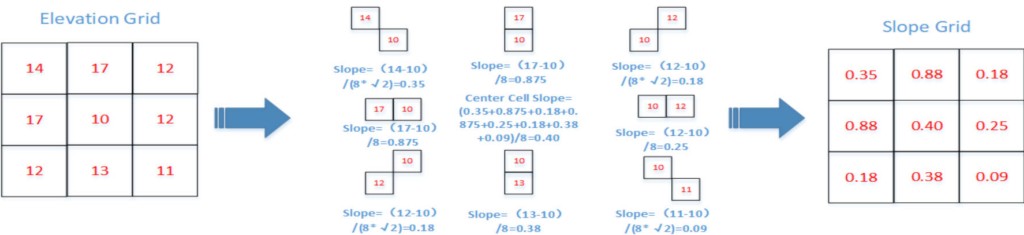

**Figure 3.** How to obtain a slope grid.

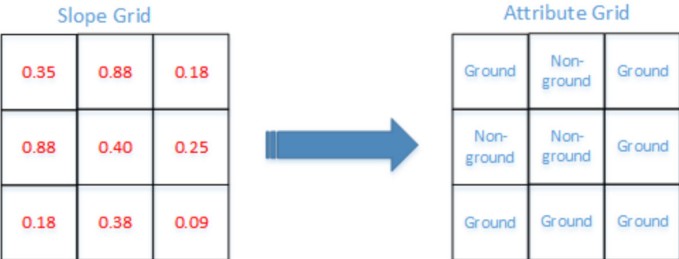

**Figure 4.** How to obtain the attribute grid.

### 2.3. Constructing a Triangular Grid and Identifying Violation Points

The Delaunay triangulation method is used to construct a triangular mesh for point clouds in this article; it has the properties of empty circumcircles and maximum–minimum angles [17]. In other words, the circumcircle of either triangle is empty for two adjacent triangles, which means that there are no points contained. In this way, the generation of elongated triangles can be avoided. This method is widely used in 3D modeling, low-poly modeling, and facial construction fields [18]. The triangular grid is constructed by this method in the present study (Figure 5). This method found that there was no elongated triangle in the internal triangular grid, except for the narrow triangle generated by cutting the point cloud area at the boundary. After generating the triangular grid, it is necessary to judge the positional relationship between the triangles in the grid, acquiring each triangle attribute in turn. If there is a common edge between two triangles, it can be considered that the two triangles are adjacent. For triangles on the boundary, if the number of adjacent triangles is less than three, the triangle is considered a boundary triangle. The computational workload in the iterative process can be reduced by disregarding the judgment violation point. Following this, the angle between the adjacent triangles is calculated in sequence. The process is as follows (see Equations (2)–(4) below):

(1) Calculate the normal vector of adjacent triangles in sequence by the calibration number of triangles ($\vec{n}$, Equation (2)), then substitute the coordinates into (2) to determine the surface normal vector ($\vec{n}$, Equation (3));

(2) Determine that the angle between the two planes is equal to the angle between the normal vectors of the two planes to obtain the angle between the two triangles ($\theta$, Equation (4));

(3) Estimate the angle and longest side length of each triangle with a threshold, and if they are more than the threshold range, the selection of a threshold angle and side length of 70° and 4 m, respectively, allows the extraction of most of the scene violation points; mark these triangles as violation triangles. Continue until all triangles are judged;

(4) Extract the maximum value of each violation triangle as violation points.

$$\vec{n} = \vec{P_1P_2} \times \vec{P_1P_3} \tag{2}$$

$$\vec{n} = \vec{P_1P_2} \times \vec{P_1P_3} = \begin{vmatrix} i & j & k \\ x_2 - x_1 & y_2 - y_1 & z_2 - z_1 \\ x_3 - x_1 & y_3 - y_1 & z_3 - z_1 \end{vmatrix} = ai + bj + ck = (a, b, c) \tag{3}$$

$$\theta = \arc \cos\left(\frac{|a_1a_2 + b_1b_2 + c_1c_2|}{\sqrt{a_1^2 + b_1^2 + c_1^2} \cdot \sqrt{a_2^2 + b_2^2 + c_2^2}}\right) \tag{4}$$

Here, $\vec{n}$ is the normal vector of the plane, $P_1, P_2, P_3$ are the three vertices of the triangle, $a$, $b$, $c$, are the three directional vectors of the triangle, and $\theta$ is the angle of two triangular.

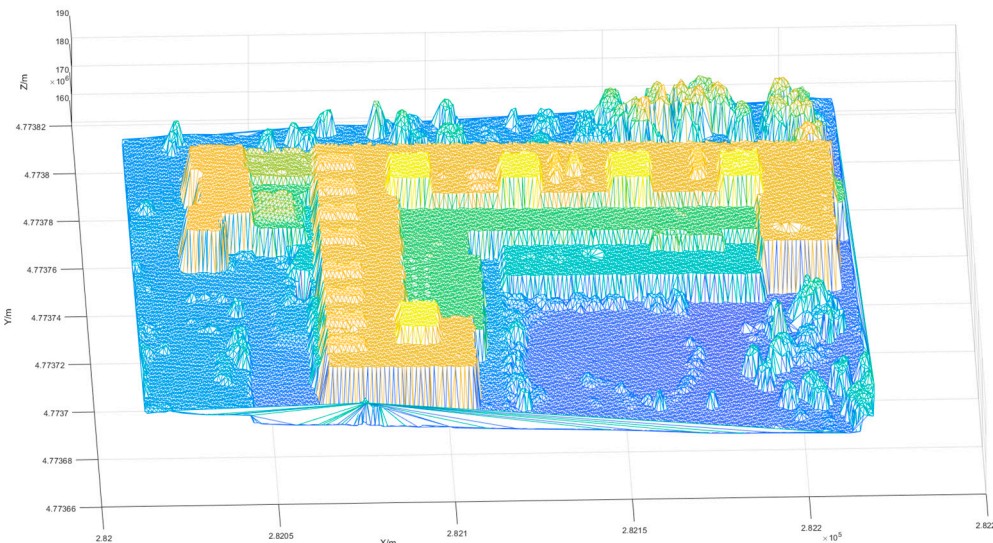

**Figure 5.** A triangular grid.

### 2.4. Collinear Judgment

The determination of the collinearity detection algorithm mainly relies on the slope between two points [19]. This method is applied to the contour extraction of buildings [20,21] and follows these steps: First, randomly select a point from the violation points as the center of the circle, and then find two other points that are closest to this point; second, calculate the slopes of two lines that connect two points to the center point and compare them to the slope between the two outliers (m, Equation (5)). If the absolute difference between these two slopes is less than the threshold, it is considered that these three points are collinear:

$$\text{m} = |(y_3 - y_1) \cdot (x_2 - x_1) - (y_2 - y_1) \cdot (x_3 - x_1)| \tag{5}$$

where m is the angle difference between two of the three points, $x_1$, $x_2$, $x_3$ are the abscissa of three points, and $y_1$, $y_2$, $y_3$ are the ordinates of three points.

### 2.5. Cluster Point Classification

The present study used KD-Tree-Based Euclidean Clustering to distinguish the violation points. The principle is that if the distance between points is less than the threshold, the two points are locally connected to the same point set. If the distance between points is greater than the distance threshold, the two points are locally connected but do not belong to the same point set (Figure 6). The process is as follows (Figure 7):

(1) Select a point from the regular violation point set, T, which is assessed by the collinear judgment as the cluster center point;

(2) Perform a neighbor point index based on the KD-Tree using the original point cloud data;

(3) Find the points within the distance threshold and add these points to the undetermined set P;

(4) Check whether the number of points in p increases or if the height difference is overrun; in which case, repeat steps 2–3 until the number of points in P does not increase;

(5) Output the P set and remove P from Q;

(6) Remove the points that are repeated with P from T to avoid repeated operations, which increase the amount of calculation;

(7) Check whether all the points in T have been calculated, and repeat steps 2–6 until there are no unoperated points.



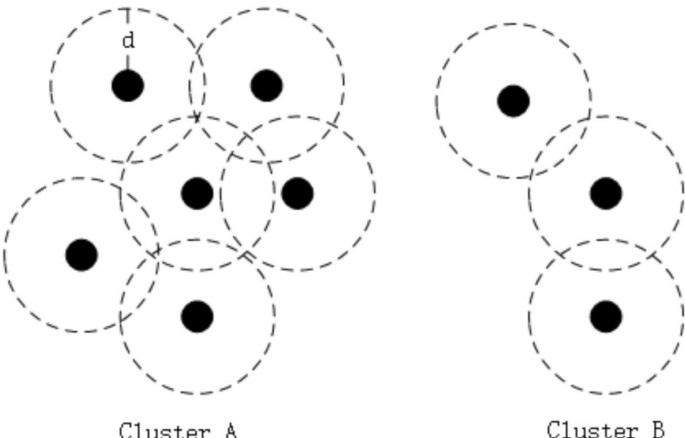

**Figure 6.** The Euclidean Clustering.

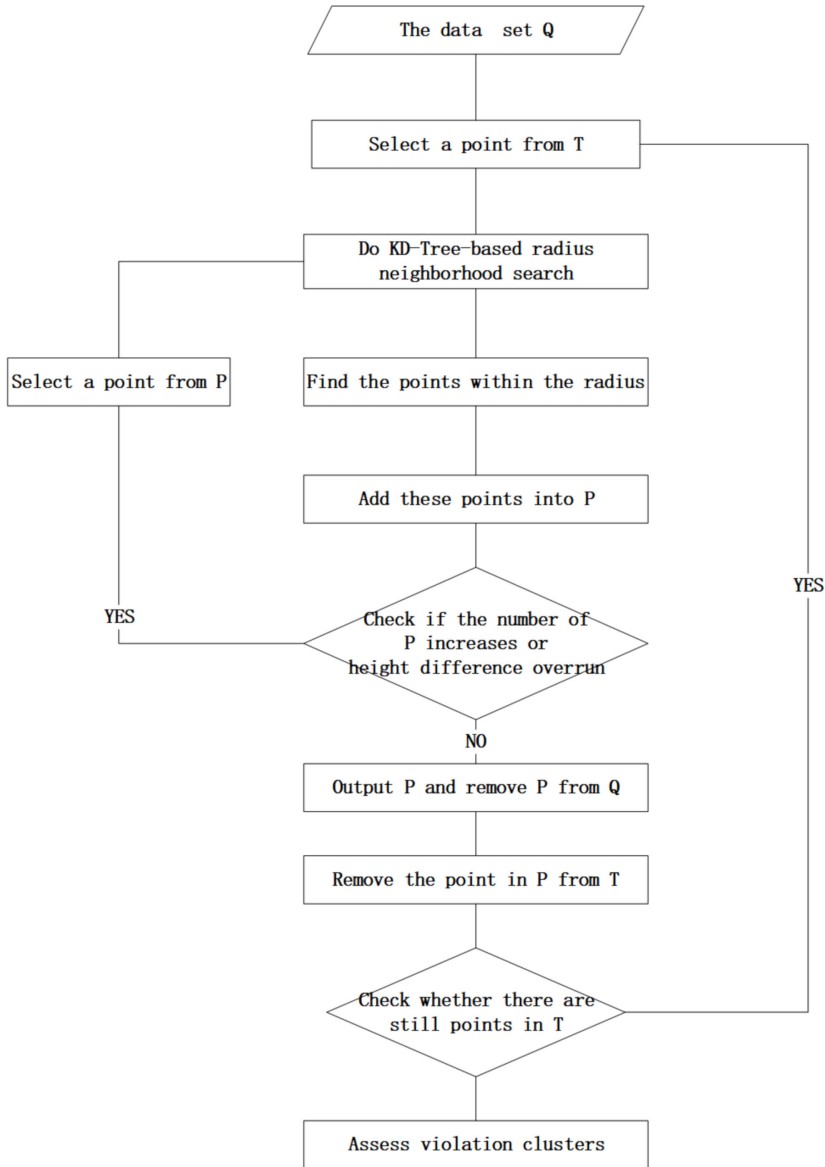

**Figure 7.** The Euclidean Clustering process.

## 3. The Experimental Procedure

### 3.1. Experimental Data and Evaluation Criteria

The study area in this article is a park in Guilin, Guangxi, China (Figure 8). The point cloud was obtained by utilizing the UAV platform. This study adopted the filter error evaluation criteria proposed by ISPRS in 2003, which include a Type I error, a Type II error, and a total error. A Type I error is the error of misclassifying ground points as non-ground points, while a Type II error is the error of misclassifying non-ground points as ground points; the total error is the ratio of the two types of errors to the total number of points in the point cloud (Table 1).

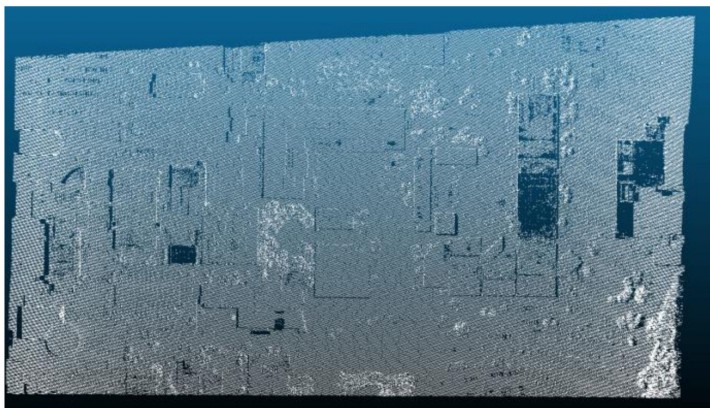

**Figure 8.** The whole study is a point cloud.

**Table 1.** Filter Error Definition.

| Reference Data | Filtered Data | | Reference Data |
|---|---|---|---|
| | **The Point of Ground** | **The Point of Non-Ground** | |
| The point of ground | a | b | e = a + b |
| The point of non-ground | c | d | f = c + d |
| The point after filter | g = a + c | h = b + d | n = a + b + c + d |

### 3.2. The Process of the Triangulation Method Filter

A representative and diverse area was selected to verify the proposed algorithm (Figure 9). The study area contained various types of scenic elements, such as dense forests, buildings, and steep slopes. The terrain in this area is uneven, and the buildings are multi-story and irregular, which means a more stringent test for the point cloud filter algorithms. Initially, a triangular grid was constructed for the study area (Figure 5). Following this, the adjacent relationships between triangles in the grid were determined, and the angles of each adjacent triangle were calculated by the cross product of vectors, comparing the angles of each adjacent triangle and the threshold. When the angle exceeded the threshold range, two triangles were marked as violation triangles at the approximate angle. On this basis, the longest edge of the common edges of the violation triangles is identified, and if the longest edge exceeds the threshold, the highest point of the triangle is marked as having the longest edge as a regular violation point (Figure 10). Based on the image of the violation points, the triangulation grid algorithm can roughly extract the outline of the buildings using the distance and angle and, subsequently, use the three-point collinearity method to determine whether the regular violation points are collinear. If they are collinear, they are marked until they traverse all regular violation points. Figure 11 shows that the outer contour of the building still has some forest points included in what are judged as regular violation points. These points will be removed in the subsequent clustering step. Using these regular violation points as centers and clustering the original point cloud by the distance of each point, this study used different colors to represent the point groups after Euclidean Clustering (Figure 12). In detail, the process requires selecting a point

from the regular violation points, taking this point as the clustering center, and using the clustering threshold to identify the cluster group of this point. Clearly, it is necessary to set an elevation threshold, and an elevation threshold of 3 m can service most scenes. In order to prevent the ground points within the threshold range from being added to the clustering group as non-ground points, after completing the clustering grouping of this point, remove the regular violation points that are in the group to avoid repeated operations, and repeat the above process until all regular points have obtained their group. Finally, by traversing the number of points in each group and retaining the group for which the number of points exceeds the set value, the steep point may be used as the violation point during extraction because the steep point is connected to the ground. The number of points after clustering is the largest, and by limiting the number of points after clustering, the steep points are avoided as non-ground points and can be removed if the number of points is less than the set value. The reasons for this situation are:

(1) When there are several separate points in the same area and the distance between them is also within the distance threshold, this situation needs to be removed because it will be easy to remove the ground points between these separate points;

(2) The points in the group may be due to the collinear situation of some forest points because the forest is roughly uneven, and this is why an elevation value is set. These groups may be composed of forest points that are misjudged as regular violation points and have forest points in close proximity.

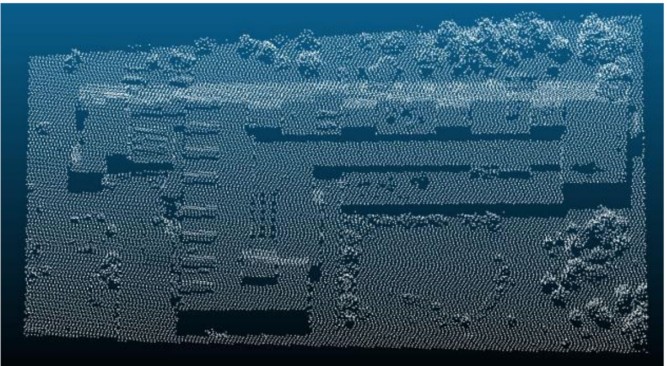

**Figure 9.** The study area's point cloud.

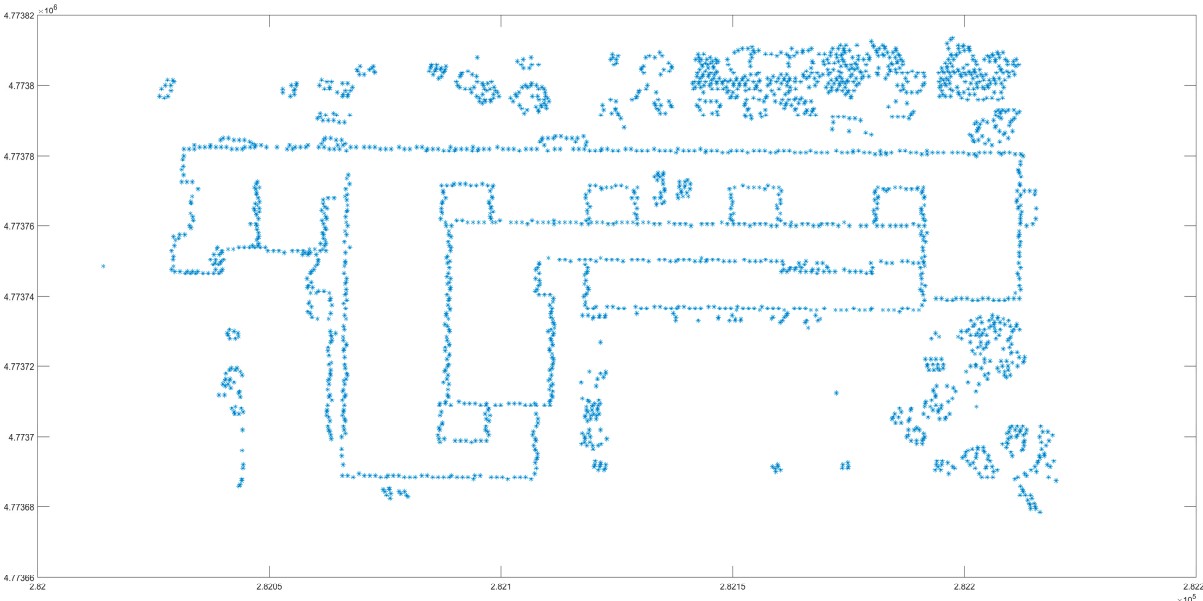

**Figure 10.** The violation points.

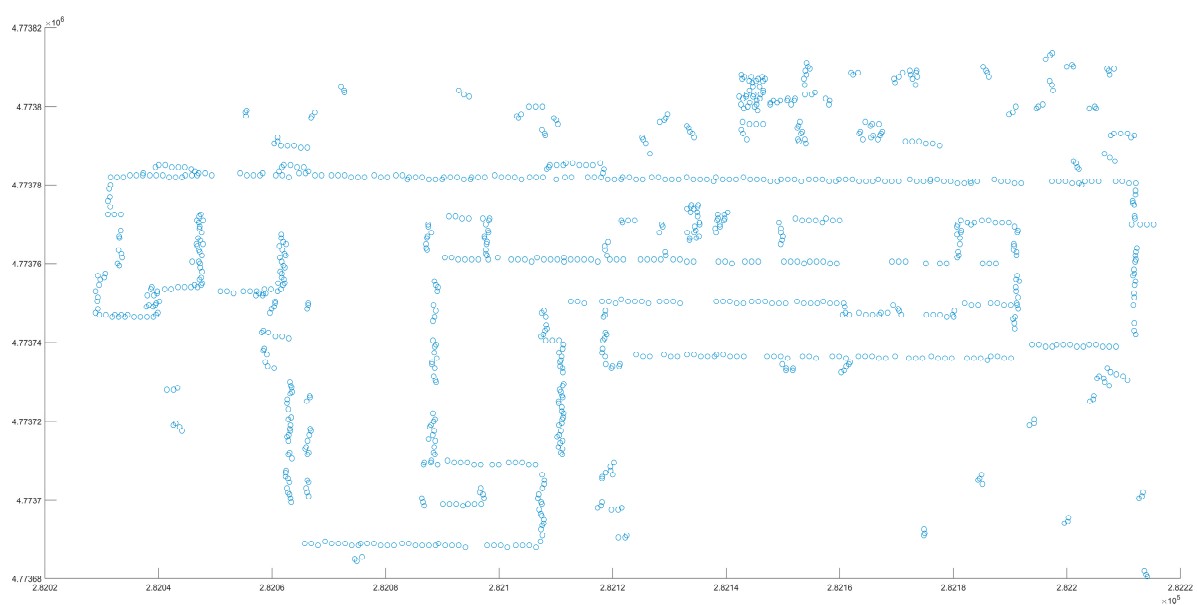

**Figure 11.** The regular violation point.

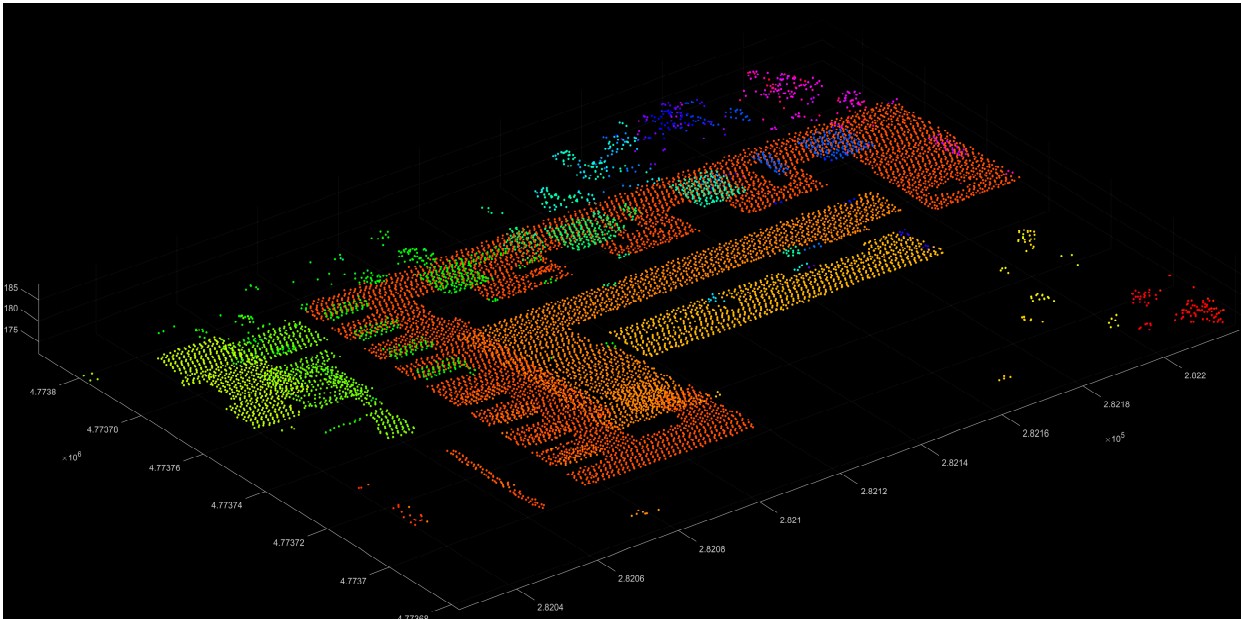

**Figure 12.** The clustering result (different color points represent different point sets after clustering).

It is necessary to establish the outer contour of the ground objects in the first non-ground point extraction process. The process is to use the outer contour to warp the non-ground objects. In this process, some non-ground points that are not identified by the clustering method are also wrapped into the non-ground point clusters, such as the points inside the house. Based on this method, the calculation amount of the second non-ground point filter is also reduced. The Convex Hull Algorithm is used to find the boundary of these clustering groups (Figure 13), and the points located within the boundary, from the original point, and these points are then removed and designated as non-ground points.

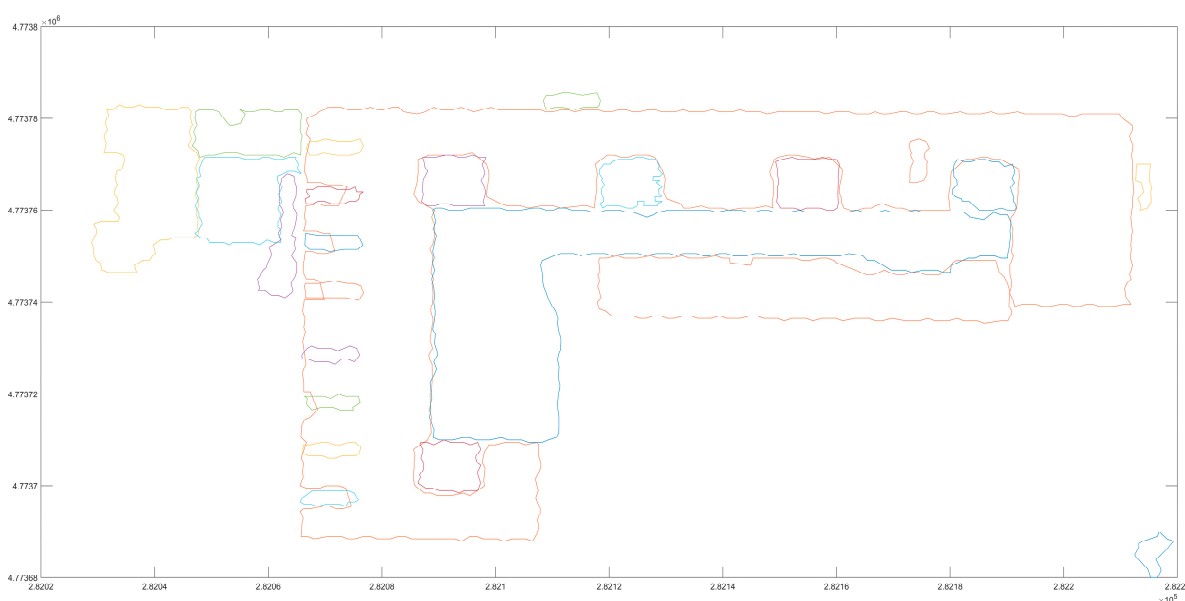

**Figure 13.** The boundary of the clustering points (different colored boundaries represent the corresponding boundaries of different point sets after clustering).

After removing some non-ground points, a new three-dimensional grid is generated, and the previous process is repeated to identify the violation points (Figure 14). However, unlike the previous process, it is necessary to reduce the screening conditions for the second-screening violation points and remove the screening condition of distance because the screening condition of distance will take the non-ground points adjacent to the ground as the ground points. In addition, the threshold of the clustering distance in the point cloud clustering process is reduced because the second round of screening is mainly aimed at scattered and irregular points; therefore, a lower clustering distance is more appropriate. After completing the above process, the purpose of separating ground points from non-ground points is achieved; the blue points in the image are the original points of the study area, and the red points are the points after the triangular grid filter. It can be seen that most of the non-ground points are separated from the original points; however, the clustering algorithm has limitations; it may misclassify some ground points that are located in clusters of non-ground points and appear as non-ground points, such as the area outlined in Figure 15. To address this issue, this study used the Slope Filter before the triangular grid filter to extract ground points. The Slope Filter classified low points with a slope attribute of ground as ground points and selected the points located at a height from the classified ground points as non-ground points. The Slope Filter has limitations and is unable to adjust its slope threshold adaptively based on terrain conditions; however, it can complement the triangular grid filter to improve the filter's efficiency and accuracy. Here we used the Slope Filter with a high threshold to extract the ground points embedded in the non-ground points (Figure 16) and the triangular grid filter to complete the separation of ground points from non-ground points (Figure 17). The blue points are the original points, and the red points are the points after being filtered. It can be seen that the method of combining two methods significantly improves the accuracy of the ground point filter. The filter effect of the marked area has also been significantly improved. Based on these results, we compared other popular filter algorithms with our method (Figures 18–23). The images that follow show the study area after the EMD Filter, and most non-ground points were removed, although three groups of non-ground points were still neglected (Figure 18). The study area after the SMRF Filter shows that most non-ground points were removed with the process of filtering; some ground points were also removed, as depicted in the blue-marked area, and some points, shown in the three red-marked areas, were neglected (Figure 19). The study area after Segmentation-Based Filtering shows that most non-ground points were removed except

for some non-ground points close to the ground; however, there are still some non-ground points that were neglected, such as the points in the red area (Figure 20). The study area after the Slope Filter shows that there are many ground points that were removed in the blue-marked area and many non-ground points that were neglected in the red-marked area; thus, we can see that the traditional Slope Filter has been unable to service most scenes, especially in the presence of composite buildings (Figure 21). The study area after Cloth Simulation Filter shows that most non-ground points were removed; however, there are still some non-ground points remaining, and with the process of the filter, many ground points located on slope scarps were removed (Figure 22). The study area after combining the Slope Filter and triangular grid filter shows that most non-ground points were removed, especially the points of the buildings, and with the help of the Slope Filter, some ground points in the forest were retained, as shown in the red-marked area (Figure 23). These images demonstrate that the triangular grid filter method based on the Slope Filter has the best effect.

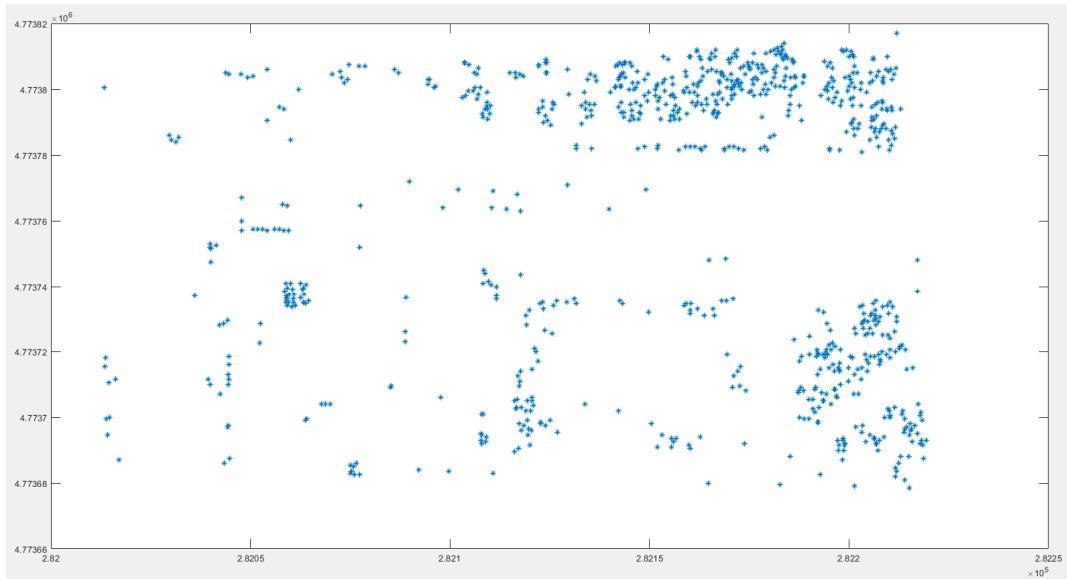

**Figure 14.** The violation points of the second round.

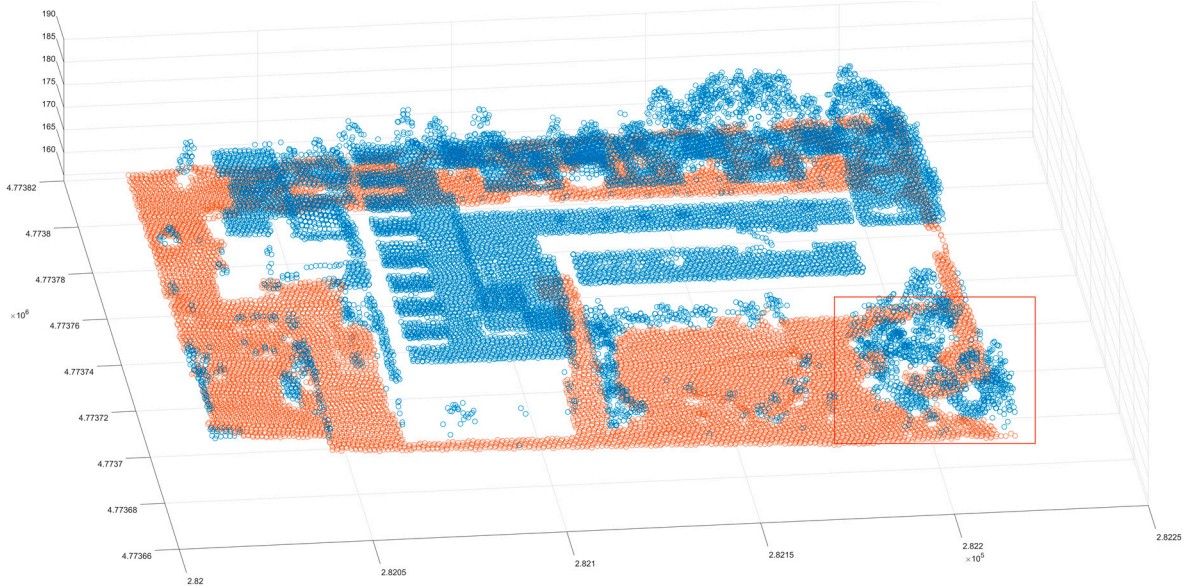

**Figure 15.** The study area after the triangular grid filter (the blue dot indicates the non-ground point, the red dot indicates the ground point, and the red box indicates the ineffective area).

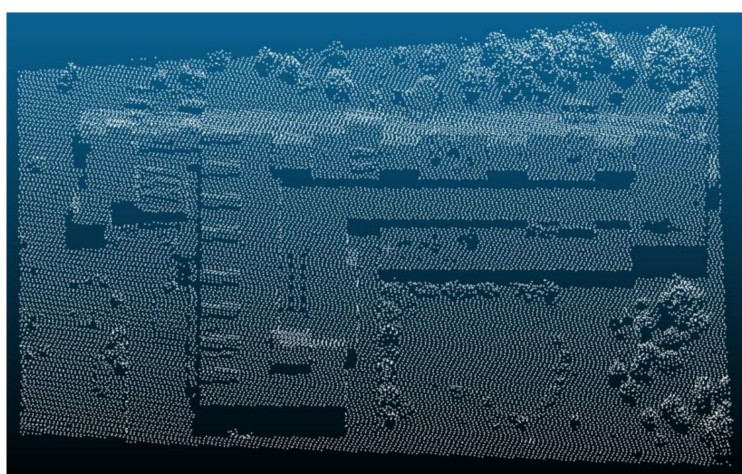

**Figure 16.** The study area after the Slope Filter with a high threshold.

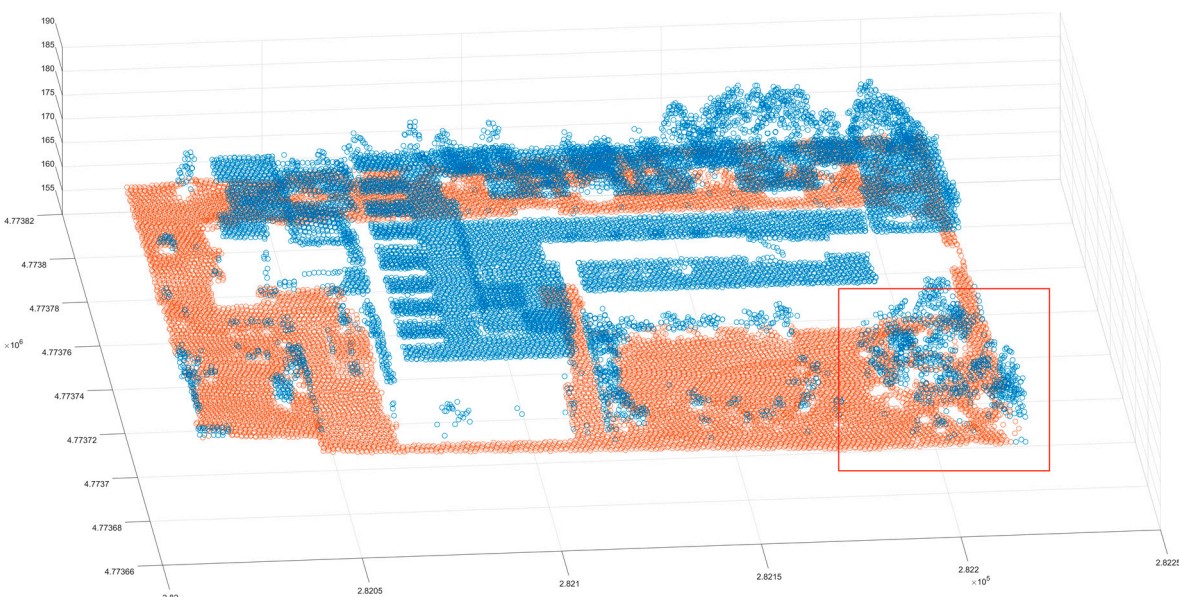

**Figure 17.** The study area after the filtering process combining two methods (the blue dot indicates the non-ground point, the red dot indicates the ground point, and the red box indicates the same area effect after the improvement).

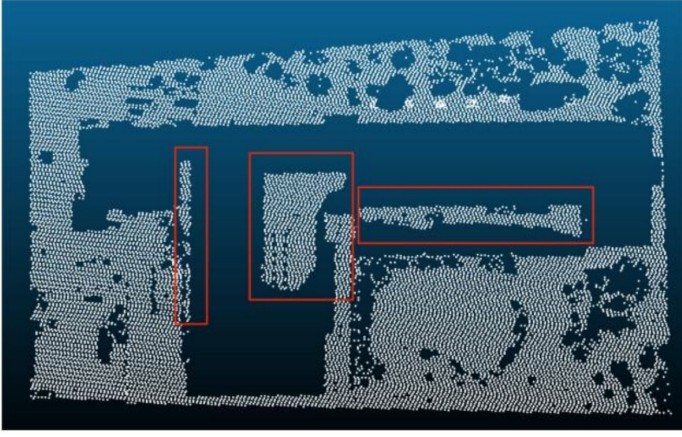

**Figure 18.** The EMD Filter (the red box selection area indicates that the non-ground point is used as the ground point retention area).

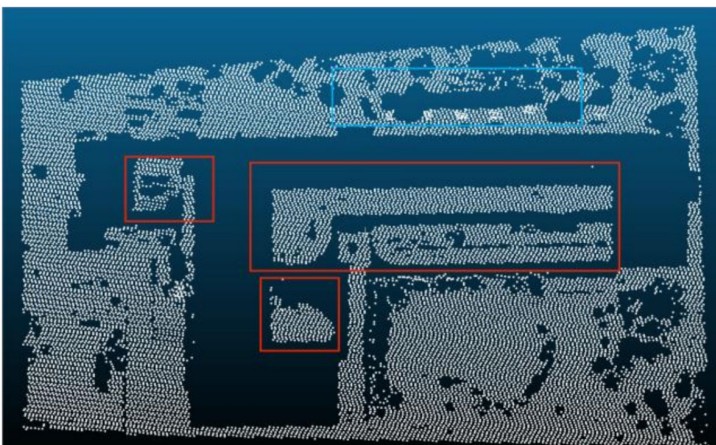

**Figure 19.** The SMRF Filter (the red box selection area indicates that the non-ground point is used as the ground point retention area).

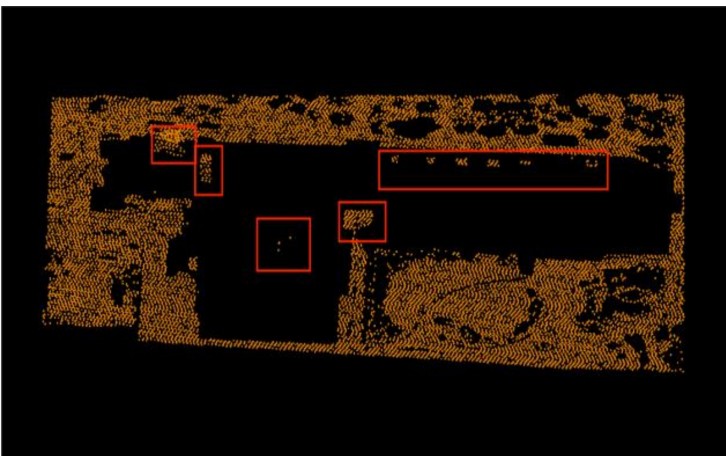

**Figure 20.** The Progressive Triangular Mesh Filter (the red box selection area indicates that the non-ground point is used as the ground point retention area).

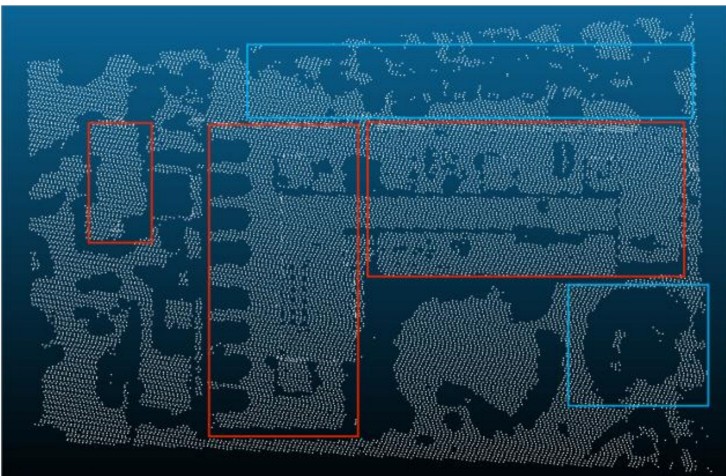

**Figure 21.** Slope Filter (the red box selection area indicates that the non-ground point is used as the ground point retention area; the blue box selection area indicates that the ground point is used as the non-ground point removal area).

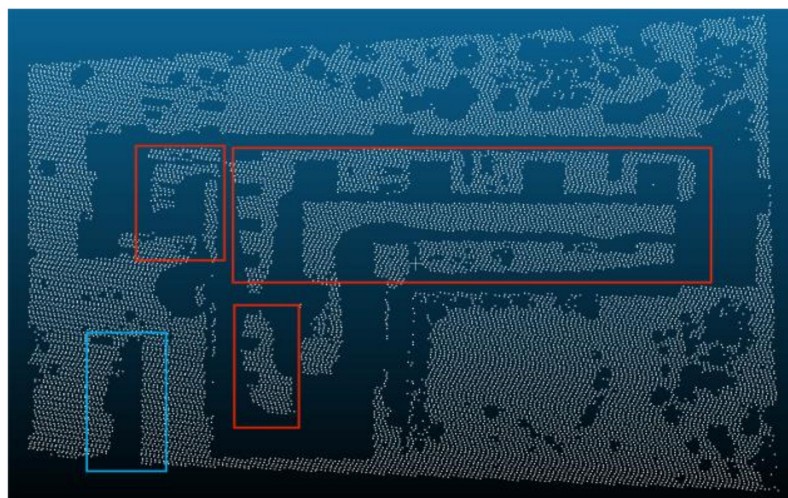

**Figure 22.** Cloth Simulation Filter (the red box selection area indicates that the non-ground point is used as the ground point retention area; the blue box selection area indicates that the ground point is used as the non-ground point removal area).

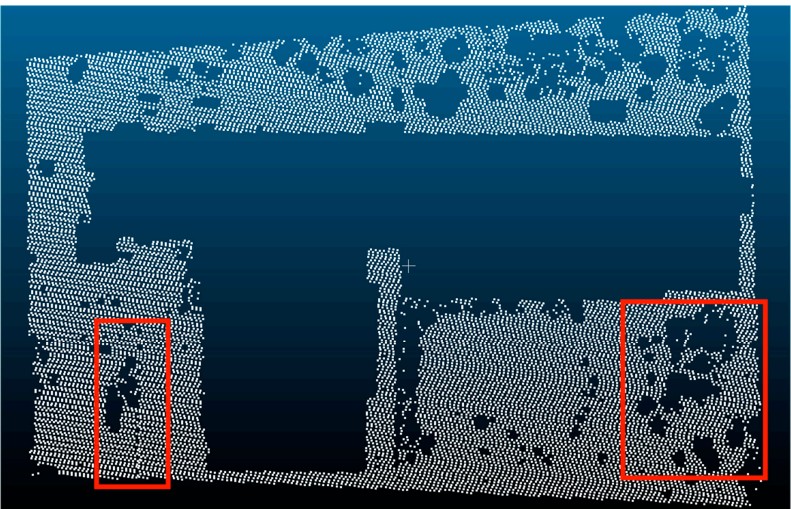

**Figure 23.** Combining the Slope Filter and triangular grid filter (the red box selected area indicates that the filtering effect is improved).

*3.3. Data Comparison*

To verify the filter's performance in this scene, this article compares the errors of the study area according to the ISPRS standards. Table 2 shows that the method used in the present study achieved better results than other methods. This is because the building in the study area has complex multi-layered structures and most filter algorithms are based on a grid filter; therefore, when facing the non-ground points on multi-layered buildings, the points on the lower layers may be misclassified as ground points, leading to a large Type I error. The Cloth Simulation Filter is affected by the cloth hardness, and this method easily misclassifies nearby buildings and ground points as non-ground points. In dense forest areas, the triangular grid filter method may remove ground points inside the convex hull boundary as non-ground points, leading to a large Type I error. The Slope Filter sets a threshold to filter out high slopes in dense forest areas, reducing the Type I error; therefore, the combination of the Slope Filter and the triangular mesh algorithm is effective in this study area.

**Table 2.** Three error types within the different methods (%).

| Filter Approach | Type I Error | Type II Error | Total Error |
|---|---|---|---|
| EMD Filter [22] | 3.5 | 33.2 | 15.4 |
| SMRF Filter [23] | 2.4 | 35.4 | 15.8 |
| Segmentation-Based Filtering [5] | 1.66 | 1.64 | 1.65 |
| Slope Filter [7] | 8.5 | 23.8 | 14.7 |
| Cloth Simulation Filter [6] | 4.57 | 2.61 | 3.77 |
| Our | 0.76 | 0.39 | 0.55 |

In order to verify the filter effect of the algorithm in other scenarios, this study used the point cloud for which the standard dataset was publicly released by ISPRS as the accuracy comparison dataset (Figures 24–38). In the relevant figures, (a) is the filtered result, and (b) is the detailed method figure. In (a), the white points are the filtered result, the blue points are the ground points marked by the original point cloud, and the red points are the non-ground points marked by the original point cloud. In (b), the white points are the ground points after filtering, the red points are Type II error points, and the blue points are Type I error points. This article calculates the Type I error, Type II error, and total error in each sample (Table 3) and compares the total error with other methods (Table 4). In sample 1-1, most non-ground points were removed, and most ground points were retained, except some ground points close to buildings because these buildings were on a hillside and close to the ground points (Figure 24). In sample 1-2, most non-ground points were removed, and most ground points were retained. Because some ground points were close to building points or vegetation points, these ground points were removed as the distance threshold was too large (Figure 25). In sample 2-1, some ground points were removed because some non-ground points were close to ground points on the boundary and the distance threshold was too large (Figure 26). In sample 2-2, some ground points were removed because they were located on the slope and the inclination angle was too large, or they were at the boundary of the scene (Figure 27). In sample 2-3, the points in the bridge and buildings were removed, and some ground points were also removed because these points were located on the slope close to non-ground points (Figure 28). In sample 2-4, most points in the buildings and vegetation were removed, and some ground points were removed because these points were close to non-ground points (Figure 29). In sample 3-1, most non-ground points were removed, and some ground points in buildings were removed because these points were close to non-ground points that belonged to low buildings (Figure 30). In sample 4-1, most buildings were removed, and some ground points were removed because these points are at the boundary of the scene and there is discontinuous terrain in the middle (Figure 31). In sample 4-2, most non-ground points were removed, and some ground points were removed because they were on the slope and close to non-ground points (Figure 32). In sample 5-1, most non-ground points were removed, and some ground points were removed because they were close to non-ground points and the distance threshold was too large (Figure 33). In sample 5-2, most non-ground points were removed, and some ground points on the ridge were removed because the angle between the points of the ridge is large and close to the non-ground points (Figure 34). In sample 5-3, most non-ground points were removed, and the ground points on the slope were also removed because the angle between the points on the discontinuous slope was too large (Figure 35). In sample 5-4, most all non-ground points were removed, and some ground points close to non-ground points were removed because the distance threshold was too large (Figure 36). In sample 6-1, most non-ground points were removed, and some non-ground points were retained because these non-ground points were close to many ground points, and if these non-ground points were removed, the filter efficiency and accuracy would be reduced (Figure 37). In sample 7-1, most non-ground points were removed, and some ground points on the slope were removed because the angle between the points on the slope was too large; these points were misjudged as non-ground points (Figure 38). By observing the filtered samples, the poor filter effect in each sample is found

and marked. The amplification of the detail shows that most of the poor filtering is due to the misclassification of ground points as non-ground points. Above all, these samples showed that the method proposed by this study can complete the point cloud filter of most scenes, but some ground points will be misjudged as non-ground points because the distance and angle thresholds cannot be adaptively changed. This, therefore, becomes the direction for improvement of the filtering methods in our proposed future research.

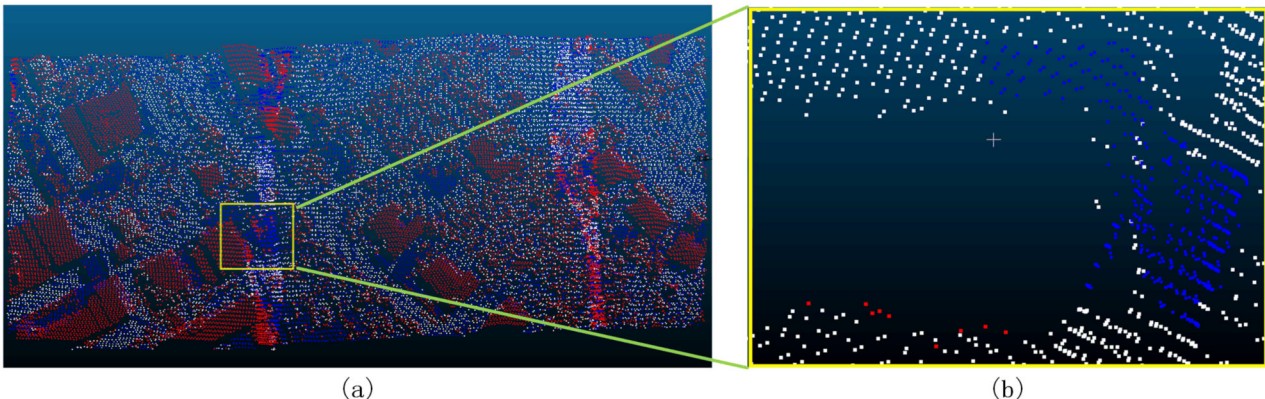

**Figure 24.** Sample 1-1 after the filter ((**a**) represents the overview of the filtering effect; the red dot in a indicates that the original data is not ground point; the blue dot indicates ground point; and the white dot indicates the remaining point after filtering as ground point; (**b**) represents the poor filtering effect; the red dot indicates Type II error; the blue dot indicates Type I error; and the white dot indicates the correct point).

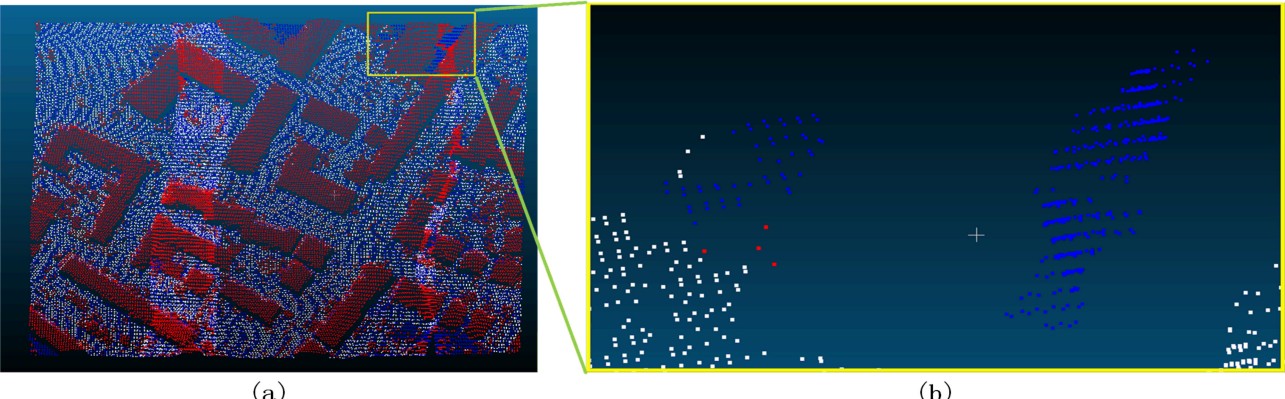

**Figure 25.** Sample 1-2 after the filter ((**a**) represents the overview of the filtering effect; the red dot in a indicates that the original data is not ground point; the blue dot indicates ground point; and the white dot indicates the remaining point after filtering as ground point; (**b**) represents the poor filtering effect; the red dot indicates Type II error; the blue dot indicates Type I error; and the white dot indicates the correct point).

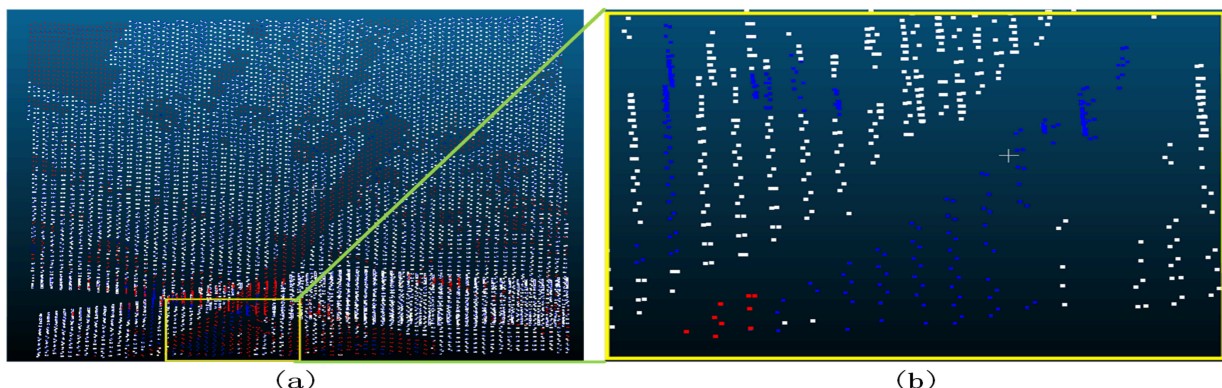

**Figure 26.** Sample 2-1 after the filter ((**a**) represents the overview of the filtering effect; the red dot in a indicates that the original data is not ground point, the blue dot indicates ground point, and the white dot indicates the remaining point after filtering as ground point; (**b**) represents the poor filtering effect; the red dot indicates Type II error, the blue dot indicates Type I error, and the white dot indicates the correct point).

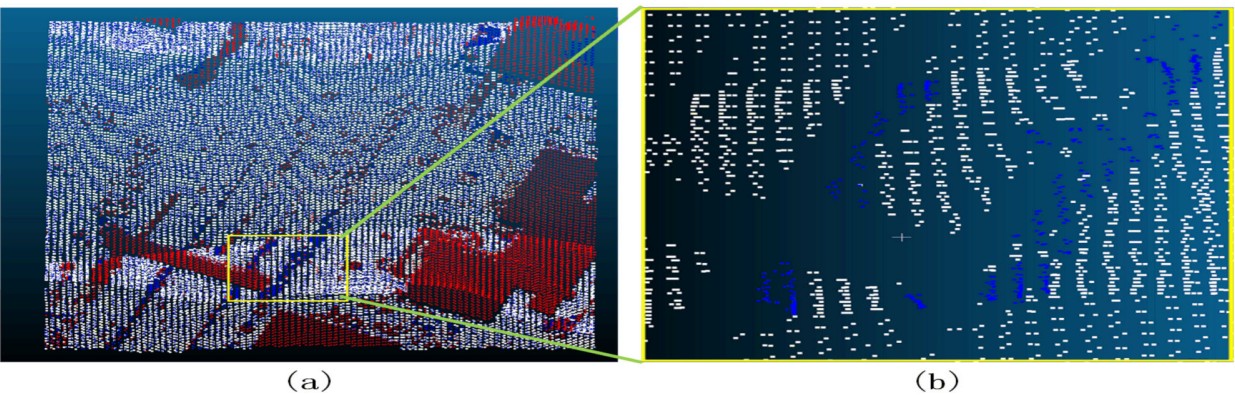

**Figure 27.** Sample 2-2 after the filter ((**a**) represents the overview of the filtering effect; the red dot in a indicates that the original data is not ground point; the blue dot indicates ground point; and the white dot indicates the remaining point after filtering as ground point; (**b**) represents the poor filtering effect; the red dot indicates Type II error; the blue dot indicates Type I error; and the white dot indicates the correct point).

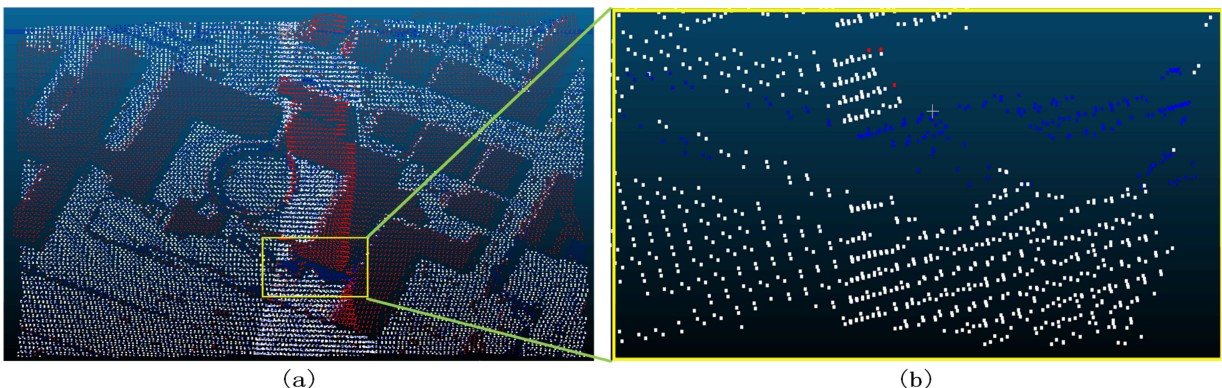

**Figure 28.** Sample 2-3 after the filter ((**a**) represents the overview of the filtering effect; the red dot in a indicates that the original data is not ground point; the blue dot indicates ground point; and the white dot indicates the remaining point after filtering as ground point; (**b**) represents the poor filtering effect; the red dot indicates Type II error; the blue dot indicates Type I error; and the white dot indicates the correct point).

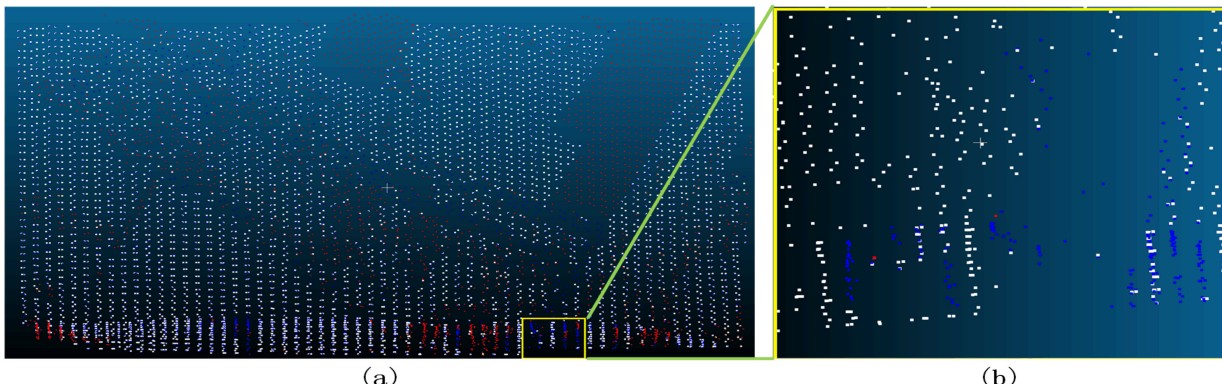

**Figure 29.** Sample 2-4 after the filter ((**a**) represents the overview of the filtering effect; the red dot in a indicates that the original data is not ground point; the blue dot indicates ground point; and the white dot indicates the remaining point after filtering as ground point; (**b**) represents the poor filtering effect; the red dot indicates Type II error; the blue dot indicates Type I error; and the white dot indicates the correct point).

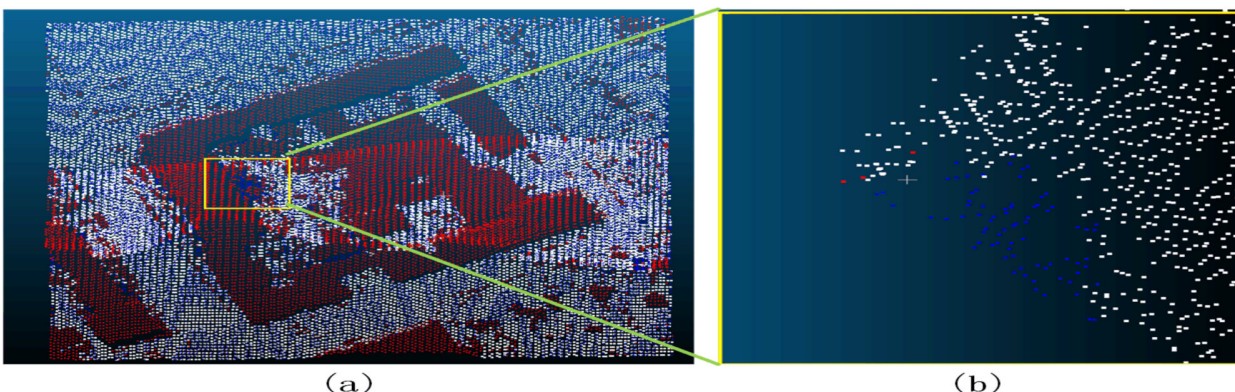

**Figure 30.** Sample 3-1 after the filter ((**a**) represents the overview of the filtering effect; the red dot in a indicates that the original data is not ground point; the blue dot indicates ground point; and the white dot indicates the remaining point after filtering as ground point; (**b**) represents the poor filtering effect; the red dot indicates Type II error; the blue dot indicates Type I error; and the white dot indicates the correct point).

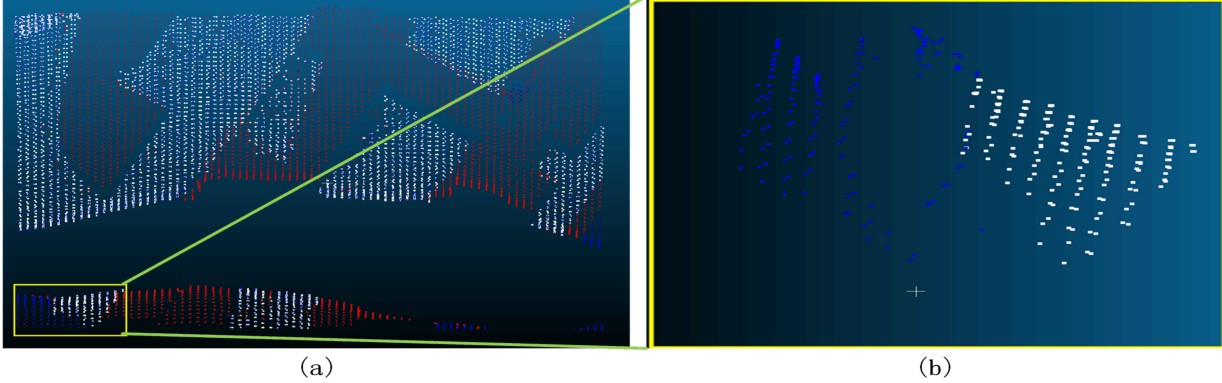

**Figure 31.** Sample 4-1 after the filter ((**a**) represents the overview of the filtering effect; the red dot in a indicates that the original data is not ground point; the blue dot indicates ground point; and the white dot indicates the remaining point after filtering as ground point; (**b**) represents the poor filtering effect; the red dot indicates Type II error; the blue dot indicates Type I error; and the white dot indicates the correct point).

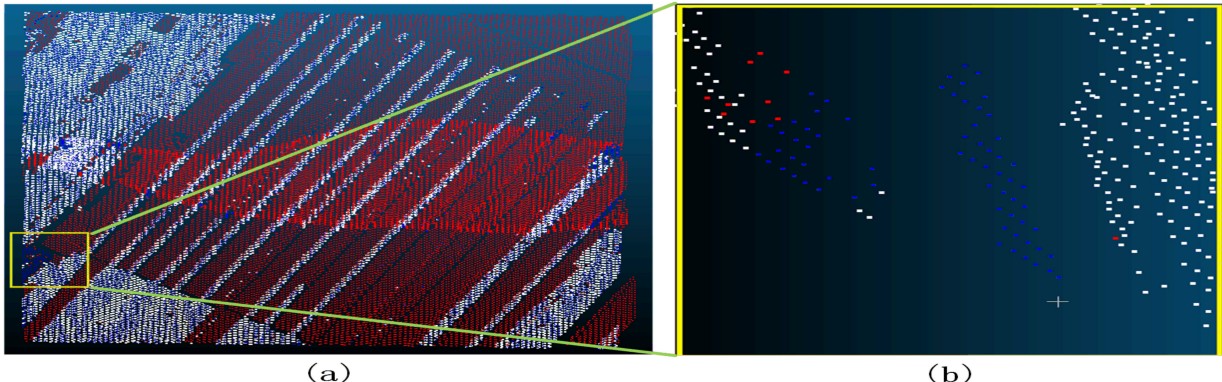

**Figure 32.** Sample 4-2 after the filter ((**a**) represents the overview of the filtering effect; the red dot in a indicates that the original data is not ground point; the blue dot indicates ground point; and the white dot indicates the remaining point after filtering as ground point; (**b**) represents the poor filtering effect; the red dot indicates Type II error; the blue dot indicates Type I error; and the white dot indicates the correct point).

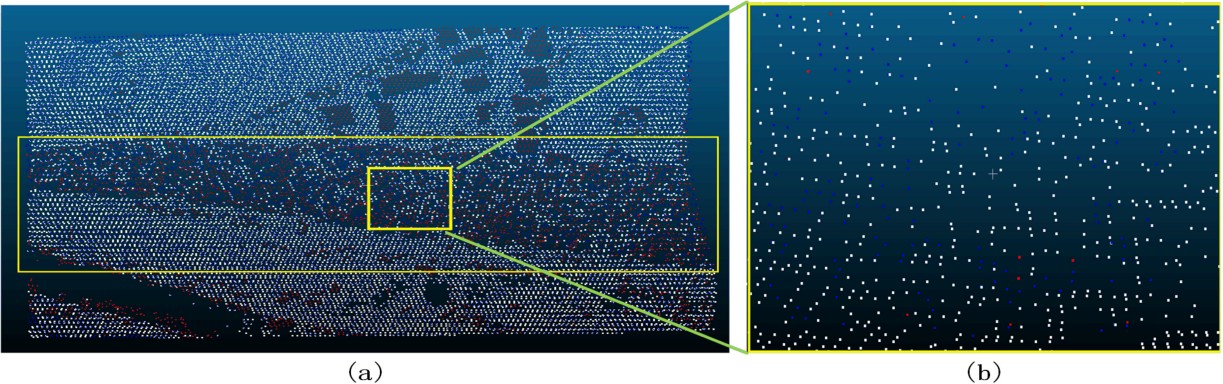

**Figure 33.** Sample 5-1 after the filter ((**a**) represents the overview of the filtering effect; the red dot in a indicates that the original data is not ground point; the blue dot indicates ground point; and the white dot indicates the remaining point after filtering as ground point; (**b**) represents the poor filtering effect; the red dot indicates Type II error; the blue dot indicates Type I error; and the white dot indicates the correct point).

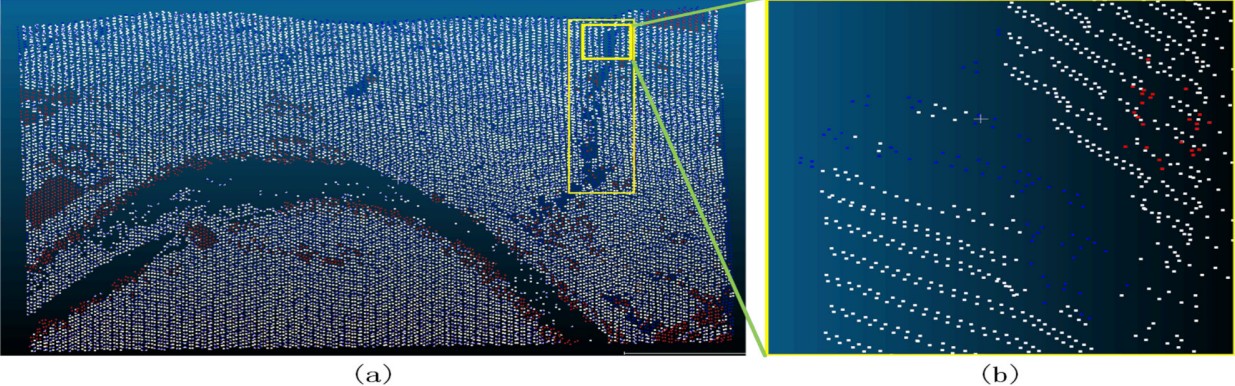

**Figure 34.** Sample 5-2 after filtering ((**a**) represents the overview of the filtering effect; the red dot in a indicates that the original data is not ground point; the blue dot indicates ground point; and the white dot indicates the remaining point after filtering as ground point; (**b**) represents the poor filtering effect; the red dot indicates Type II error; the blue dot indicates Type I error; and the white dot indicates the correct point).

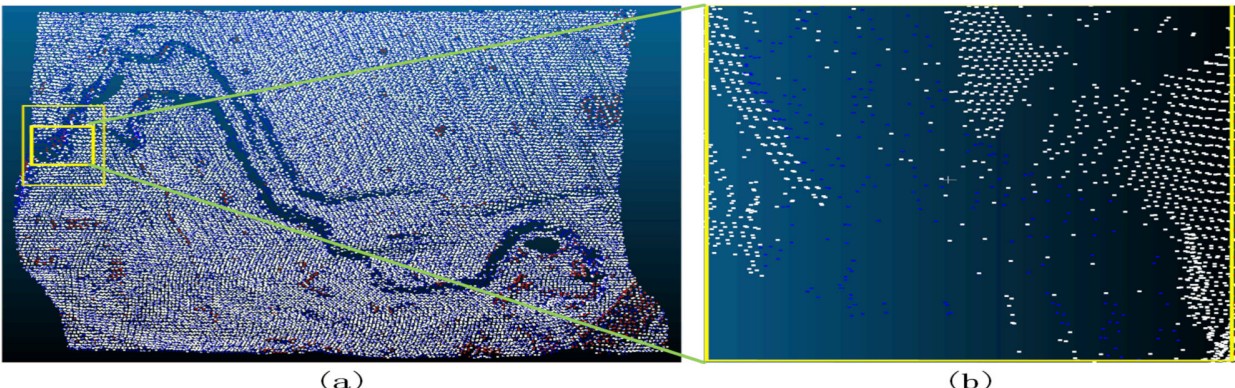

**Figure 35.** Sample 5-3 after the filter ((**a**) represents the overview of the filtering effect; the red dot in a indicates that the original data is not ground point; the blue dot indicates ground point; and the white dot indicates the remaining point after filtering as ground point; (**b**) represents the poor filtering effect; the red dot indicates Type II error; the blue dot indicates Type I error; and the white dot indicates the correct point).

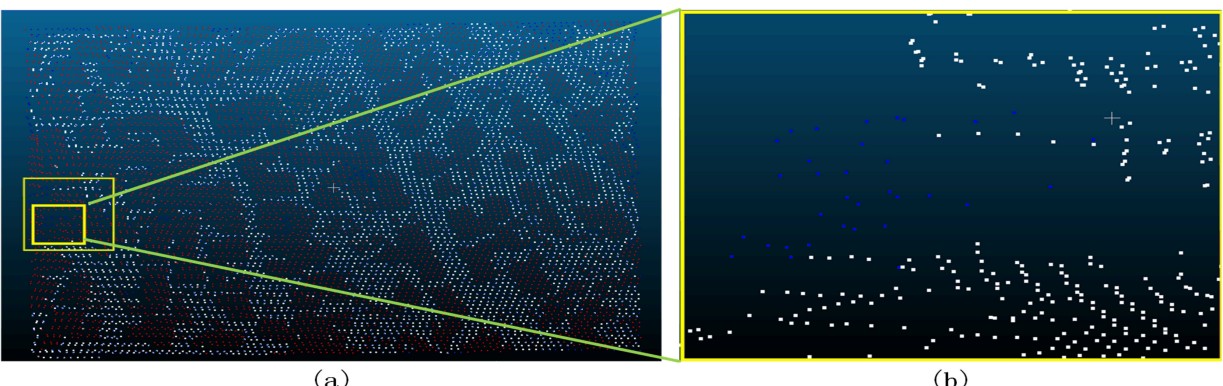

**Figure 36.** Sample 5-4 after the filter ((**a**) represents the overview of the filtering effect; the red dot in a indicates that the original data is not ground point; the blue dot indicates ground point; and the white dot indicates the remaining point after filtering as ground point; (**b**) represents the poor filtering effect; the red dot indicates Type II error; the blue dot indicates Type I error; and the white dot indicates the correct point).

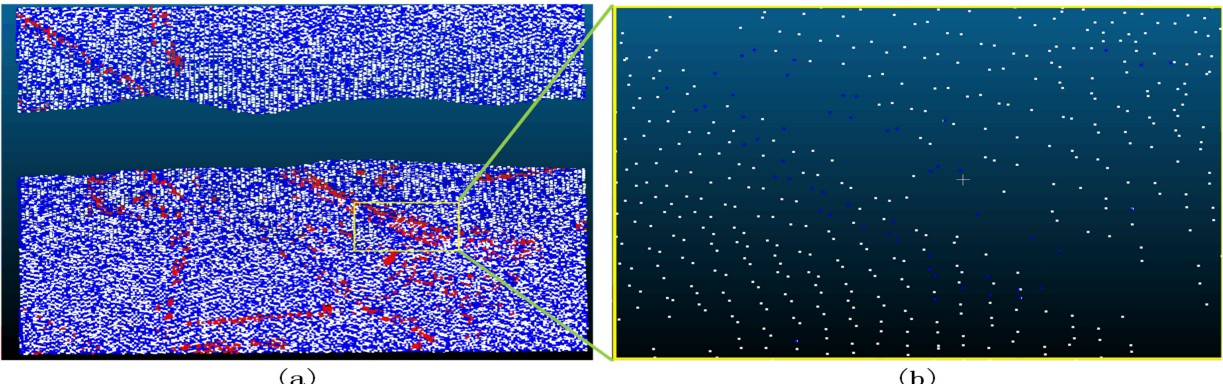

**Figure 37.** Sample 6-1 after the filter ((**a**) represents the overview of the filtering effect; the red dot in a indicates that the original data is not ground point; the blue dot indicates ground point; and the white dot indicates the remaining point after filtering as ground point; (**b**) represents the poor filtering effect; the red dot indicates Type II error; the blue dot indicates Type I error; and the white dot indicates the correct point).

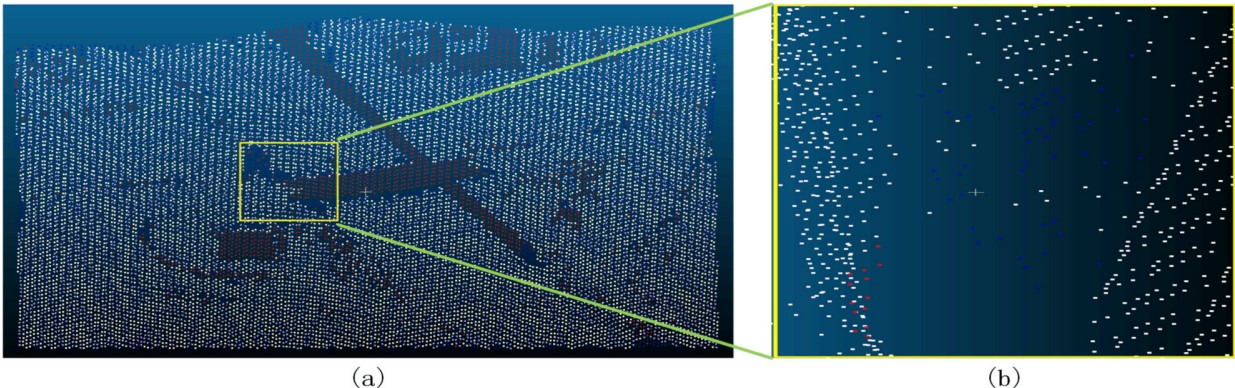

(a)           (b)

**Figure 38.** Sample 7-1 after the filter ((**a**) represents the overview of the filtering effect; the red dot in a indicates that the original data is not ground point; the blue dot indicates ground point; and the white dot indicates the remaining point after filtering as ground point; (**b**) represents the poor filtering effect; the red dot indicates Type II error; the blue dot indicates Type I error; and the white dot indicates the correct point).

**Table 3.** Three error types within the different samples (%).

| Sample | Type I Error | Type II Error | Total Error |
|---|---|---|---|
| 1-1 | 10.76 | 3.86 | 7.82 |
| 1-2 | 4.68 | 2.32 | 2.81 |
| 2-1 | 2.70 | 1.57 | 2.45 |
| 2-2 | 2.10 | 0.72 | 1.94 |
| 2-3 | 3.32 | 2.14 | 2.76 |
| 2-4 | 5.26 | 5.15 | 5.23 |
| 3-1 | 1.90 | 1.87 | 1.88 |
| 4-1 | 10.64 | 0.98 | 5.80 |
| 4-2 | 3.76 | 0.26 | 1.29 |
| 5-1 | 5.74 | 2.93 | 5.13 |
| 5-2 | 2.14 | 4.91 | 2.43 |
| 5-3 | 2.41 | 23.47 | 3.26 |
| 5-4 | 5.72 | 5.15 | 5.41 |
| 6-1 | 1.05 | 31.76 | 2.57 |
| 7-1 | 1.77 | 25.59 | 4.46 |
| average | 4.26 | 7.51 | 3.68 |

**Table 4.** The total error of the different methods (%).

| Site | Sample | Axelsson [5] | Pfeifer [24] | Sohn [25] | Elmqvist [26] | Roggero [27] | Brovelli [28] | Sithole [2] | Wack [29] | Wang [9] | Zhu [8] | Our |
|---|---|---|---|---|---|---|---|---|---|---|---|---|
| Urban | 1-1 | 10.76 | 17.35 | 20.49 | 22.40 | 20.80 | 36.96 | 23.25 | 24.02 | 17.74 | 14.87 | 7.82 |
| | 1-2 | 3.25 | 4.50 | 8.39 | 8.18 | 6.61 | 16.28 | 10.21 | 6.61 | 5.34 | 3.14 | 2.81 |
| | 2-1 | 4.25 | 2.57 | 8.8 | 8.53 | 9.84 | 9.30 | 7.76 | 4.55 | 4.90 | 3.63 | 2.45 |
| | 2-2 | 3.63 | 6.71 | 7.54 | 8.93 | 23.78 | 22.28 | 20.86 | 7.51 | 8.17 | 5.92 | 1.94 |
| | 2-3 | 4.00 | 8.22 | 9.84 | 12.28 | 23.20 | 27.80 | 22.71 | 10.97 | 8.50 | 12.34 | 2.76 |
| | 2-4 | 4.42 | 8.64 | 13.33 | 13.83 | 23.25 | 36.06 | 25.28 | 11.53 | 8.75 | 8.36 | 5.23 |
| | 3-1 | 4.78 | 1.80 | 6.39 | 5.34 | 2.14 | 12.92 | 3.15 | 2.21 | 4.93 | 4.74 | 1.88 |
| | 4-1 | 13.91 | 10.75 | 11.27 | 8.76 | 12.21 | 17.03 | 23.67 | 9.01 | 7.91 | 11.44 | 5.80 |
| | 4-2 | 1.62 | 2.64 | 1.78 | 3.68 | 4.30 | 6.38 | 3.85 | 3.54 | 3.48 | 3.30 | 1.29 |
| Rural | 5-1 | 2.72 | 3.71 | 9.31 | 21.31 | 3.01 | 22.81 | 7.02 | 11.45 | 7.05 | 4.61 | 5.13 |
| | 5-2 | 3.07 | 19.64 | 12.04 | 57.95 | 9.78 | 45.56 | 27.53 | 23.83 | 6.10 | 4.89 | 2.43 |
| | 5-3 | 8.91 | 12.60 | 20.19 | 48.45 | 17.29 | 52.81 | 37.07 | 27.24 | 4.33 | 7.71 | 3.26 |
| | 5-4 | 3.23 | 5.47 | 5.68 | 21.26 | 4.96 | 23.89 | 6.33 | 7.63 | 5.57 | 3.90 | 5.41 |
| | 6-1 | 2.08 | 6.91 | 2.99 | 35.87 | 18.99 | 21.68 | 21.63 | 13.47 | 3.26 | 2.01 | 2.57 |
| | 7-1 | 1.63 | 8.85 | 2.20 | 34.22 | 5.11 | 34.98 | 21.83 | 16.97 | 7.56 | 4.21 | 4.46 |
| average | | 4.82 | 8.02 | 9.35 | 20.73 | 12.34 | 25.78 | 17.48 | 12.04 | 6.91 | 6.34 | 3.68 |

The comparison of the different methods shows that the Type I, Type II, and total errors of the method proposed in this article are much smaller than the other filter methods applied to the study area, and that the proposed filter effect is the best (Table 2). In 15 samples, the method proposed in this article can remove most of the non-ground points, but it will inevitably remove some ground points in the filter process, resulting in large Type I errors. In addition, we found that the average of the Type II errors was higher than that of the Type I errors (Table 3). The main reason for this is that, in the filtering process, there were several samples with a small number of non-ground points; therefore, even if there were fewer non-ground points that were not identified in the filtering process, it would lead to a large type II error. However, from the overall situation comparison, the Type I error in most samples is still greater than the Type II error. Comparing with the filter methods proposed by other scholars, our method outperforms other methods in most scenarios, and the average error of 3.68% is smaller than in the other methods (Table 4). In order to intuitively see the errors between the different methods, we compiled the error comparison diagram for each of the samples (Figure 39). The results show that the error in these samples is less than in other methods, except in four samples where there are a large number of near-ground points, which cause a higher Type I error because the clustering threshold cannot be adaptively changed. Further analysis of the results shows that our method works well in urban samples, and compared with other methods, the total error is the smallest in urban samples, except for sample 2-4. In addition, while the total error of our method is not the smallest in the three rural samples, the filter effect is reasonable, and there is evidently a certain effect on the extraction of large scene buildings. The extraction effect of scattered points and small objects is also good. From the range of curve changes in each scene, we can see that compared with other methods, the error amplitude of this method fluctuates very little, the filter effect of each sample is relatively stable, and the error is controlled within 10%, showing that this method has strong adaptability and can provide an effective filter for point cloud data in multiple scenes.

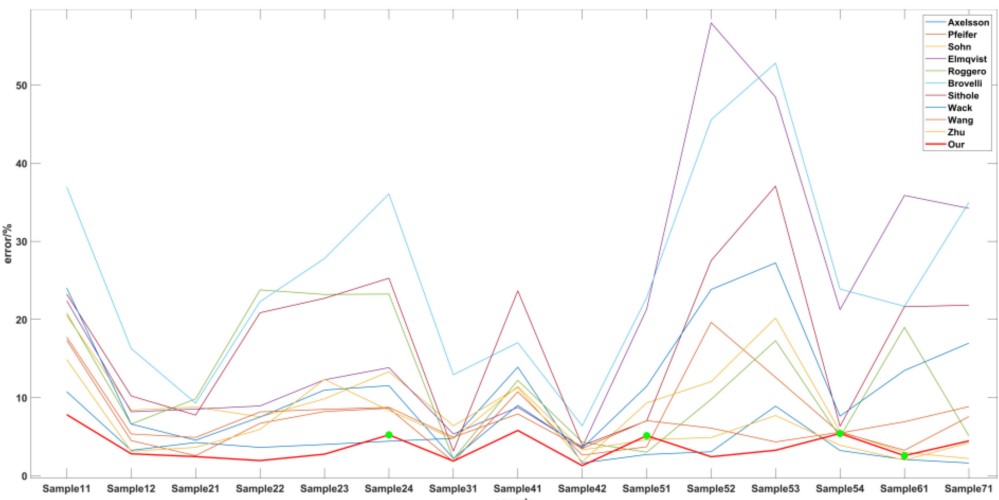

**Figure 39.** The error of different methods in each sample.

In order to more intuitively observe whether the ground point cloud filter can be applied afterward in practice, this study selected a representative sample from the urban and rural samples as the verification and used the original data (Figures 40a and 41a), the ground data in the original data (Figures 40b and 41b), and the filter data (Figures 40c and 41c) to make the surface reconstruction model. The results show that the model generated by the filter points removed most non-ground objects and retained the geomorphological features, so that the generated terrain model is almost the same as the model composed of ground points. The results confirm that the filter method reported in this article has a certain utility and can meet the requirements of the point cloud filter.

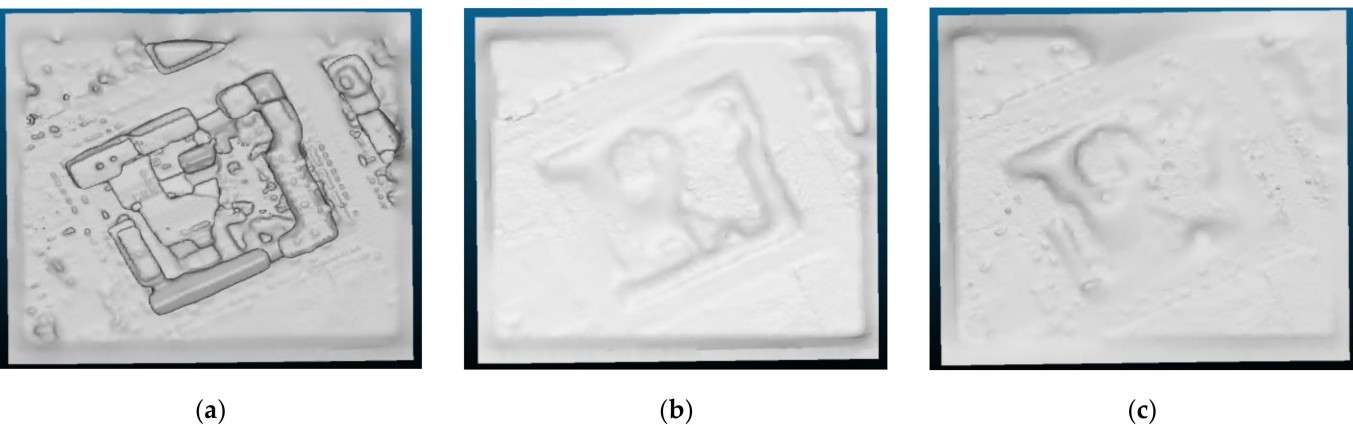

<div align="center">(<b>a</b>)        (<b>b</b>)        (<b>c</b>)</div>

**Figure 40.** Sample 3-1 surface reconstruction ((**a**) represents the original point cloud to construct the surface model; (**b**) represents the original ground point cloud to construct the surface model; (**c**) represents the filtered ground point to construct the surface model).

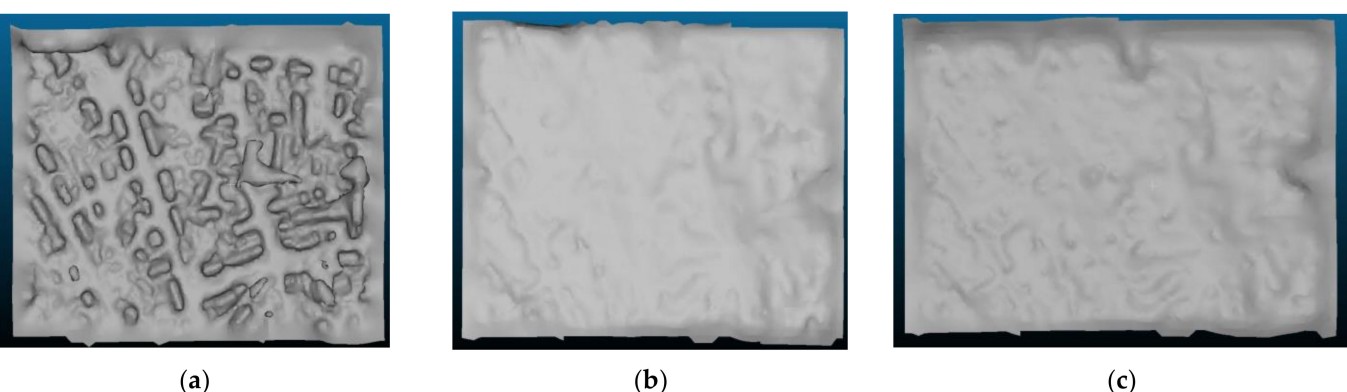

<div align="center">(<b>a</b>)        (<b>b</b>)        (<b>c</b>)</div>

**Figure 41.** Sample 5-4 surface reconstruction ((**a**) represents the original point cloud to construct the surface model; (**b**) represents the original ground point cloud to construct the surface model; (**c**) represents the filtered ground point to construct the surface model).

## 4. Discussion

As an effective tool of the point cloud filter, the Slope Filter has been widely used in the classification of ground points and non-ground points, dividing the study area into grids, which is the first key step in this method. The ground grids and non-ground grids are distinguished by these grids, and the slope is calculated separately to determine the lower points as the ground points. This article proposed a filter method based on the Slope Filter that obtained the adjacent state between each point, determined the distances and angles of each point, and calculated the violation points. The principle is that there must be a certain violation area between the non-ground area and the ground area. Simply speaking, there will be an unnatural angle and distance area, unlike the ground slope, which will rise slowly at a certain natural angle. Therefore, this article establishes a triangular grid to obtain the violation area, selecting two adjacent triangles in the grid. When the angle of these planes is too large and the length of any edge in the two planes exceeds the threshold, it can be determined that the triangle with the longest edge is the unnatural area; thus, the violation point is identified from this triangle.

An airborne radar usually obtains the point cloud data from a large area. The characteristics of this method are that the horizontal distribution of the points presents a certain distance distribution. It can be thought that if there is no contact with other objects in scanning, the points of the same object often appear as cluster points. Based on this idea, this article adds the KD-Tree-Based Clustering algorithm to classify the objects in the scene. This method can express the distance relationship between each point and is applied to the

extraction of the building point cloud; however, there are still some problems with the point cloud filter. When the point cloud has near-ground scattered points, such as in a forest, it is easy to eliminate the ground points near the scattered points as non-ground points using a large clustering threshold. Therefore, this article proposes the use of a large threshold to extract higher non-ground points; most of these non-ground points are buildings. This article used a three-point collinear method because these violation points show a regular distribution and can be clustered and eliminated using the lower threshold to eliminate the scatter points. In the process of airborne radar scanning, it is inevitable to scan the building façade; therefore, this study added a height threshold to avoid the mutual conduction between the points in the clustering process and the elimination of the ground points.

Compared with using the grid, the triangular grid can more accurately express the relationship between each point; it can filter non-ground points when facing multiple buildings with different heights; and it can determine whether the points are ground points in the face of discontinuous areas. This method is accurate, efficient, and can be applied to a variety of scenarios. The total error of the point cloud filter algorithm is only 3.68%, which is smaller than most algorithms, and the error source is mainly due to a Type I error, the source of which can be explained as follows: (1) In the process of judging the violation point, the ground points are in the area with a significant change in slope; therefore, these points were judged as violation points. However, with the development of LiDAR technology, the density and quality of the point cloud will be improved, and the subsequent misjudgment of the violation point will be reduced. (2) Due to the limitations of the clustering algorithm, some ground points close to non-ground points will be judged as non-ground points. This is illustrated by the process of calculating the sample data from ISPRS, where there is a situation where the Type I error is too large. The reason for this is that the number of non-ground points in the sample data is too small, and if using this method repeatedly to separate the point cloud, the total error will be too large. However, in order to confirm the practicability of the algorithm in a variety of areas, the Type II error is reduced as much as possible. In summary, this algorithm provided a new idea for the point cloud filter, and its efficiency can be improved with the development of the point cloud quality and clustering algorithms.

The research content of this article still has limitations:

(1) The clustering algorithm is still not perfect, as it fails to adjust the distance threshold based on the point cloud distribution in the relevant scene being studied;

(2) This method needs to construct a triangular grid for each point in the point cloud, so the processing speed of point cloud data with large scenes is relatively slow, and complex scenes require repeated operations;

(3) The filter effect is poor when the slope on discontinuous ground changes significantly.

## 5. Conclusions

This article proposes a novel and efficient triangular grid algorithm based on the Slope Filter that can be applied to a variety of areas. The establishment of a triangular grid more effectively expresses the spatial attributes of each point, and the use of the clustering algorithm to achieve the transformation from the violation point to the violation cluster point improves the operation efficiency of the point cloud filter process. This article verified that the clustering algorithm can be applied to point cloud filtering, and through the comparison of multiple sets of experiments, this method achieves good results with multi-story buildings. In the experimental comparison of the sample data provided by ISPRS, this method showed better stability and accuracy than other methods, and this article subsequently verifies its theoretical practicability using the point cloud characteristics.

Our method has the following aspects worthy of future research and improvement:

(1) The clustering algorithm used a fixed threshold when obtaining non-ground points; however, it is easy to misclassify ground points that are closer to non-ground points. Our future research proposes a clustering method that can adopt an adaptive threshold;

(2) After each round of point cloud filtering, it is necessary to rebuild the grid, which reduces the efficiency of some operations. Our future research proposes the gradual optimization of the present algorithm model to further enhance its computational efficiency;

(3) For scenes with more scatter distribution, the filtering effect may be poor. In the future, we will further optimize the scatter scene model to enhance the accuracy of the operation.

**Author Contributions:** Supervision, C.K., S.W., Y.L., C.G. and S.Z.; Writing—original draft, Z.L.; Writing—review and editing, C.K. and Z.L. All authors have read and agreed to the published version of the manuscript.

**Funding:** This study was supported by the National Natural Science Foundation of China (41961063, 42064002).

**Data Availability Statement:** Not applicable.

**Acknowledgments:** The authors would like to thank the editors and the reviewers for their valuable suggestions.

**Conflicts of Interest:** The authors declare no conflict of interest.

## Abbreviations

The following abbreviations are used in this manuscript:

| | |
|---|---|
| ISPRS | International Society for Photogrammetry and Remote Sensing |
| EMD | Empirical Mode Decomposition |
| SMRF | Simple Morphological Filter |

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
