# Peer review of "A Triangular Grid Filter Method Based on the Slope Filter"

_remotesensing, doi:10.3390/rs15112930_

Round 1

Reviewer 1 Report (Previous Reviewer 1)

Community Verified icon The author proposes a triangular mesh filter based on the slope filter, which has been verified by experiments to have good accuracy and stability, and has achieved good results in separating grounded and non-grounded points. I believe the algorithm is well innovative and well validated in experiments. I believe the revised English grammar can be published in RS. Community Verified icon English can be published with some modifications.

Author Response

Thank you very much for your affirmation of the content of our article. We will carefully modify some English language problems in the article. In the subsequent submission process, we will also invite a professional English polishing team to further polish and modify our article. Once again, thank you very much for your comments and suggestions.

Reviewer 2 Report (Previous Reviewer 2)

All comments have done

Author Response

Thank you very much for your affirmation of the content of our article. Best wishes for you, and wish success for you and your team.

Reviewer 3 Report (New Reviewer)

The manuscript presents a method for filtering lidar data to eliminate non-terrain points.

The method is explained clearly enough.

The results obtained show superior performances compared to the most used methods.

However, this was a predictable result, given that the procedure described is, in reality, a mixture of several techniques.

Author Response

Thank you very much for your affirmation of the content of our article. Although the method proposed in this paper is to mix several methods, we also make some modifications to the characteristics of point cloud scene in these methods. Most of the filtering methods can not deal with the problem of multi-layer building point cloud scene, this method can get good results for this scenario, and through this study, it is verified that the clustering method can be applied to point cloud filtering, and can get good filtering effect. Compared with other methods, the efficiency and accuracy have been improved. Once again, thank you very much for your comments and suggestions.

Reviewer 4 Report (New Reviewer)

The authors must also clearly discuss the significance of the research problem in the first section.

Please give a frank account of the strengths and weaknesses of the proposed research method. 

In the conclusion section, the limitations of the proposed method must be discussed by the authors 

 In the conclusion section, the authors will need to clearly address their research contributions in theory

Author Response

Response to Reviewer 4 Comments

We feel great thanks for your professional review work on our article. As you are concerned, there are several problems that need to be addressed. According to your nice suggestions, we have made extensive corrections to our previous draft, the detailed corrections are listed below.

Point 1: The authors must also clearly discuss the significance of the research problem in the first section.

Response 1: Thanks for your suggestion, on page 3, line 101, we have modified the introduction part, briefly described the limitations of the current filtering method, highlighted the necessity of the proposed method, and explained the significance of using this method.

Point 2: Please give a frank account of the strengths and weaknesses of the proposed research method. 

Response 2: We sincerely appreciate the valuable comments, on page 23, line 537, We illustrate some limitations of this method, mainly reflected in three aspects :

  1. The clustering threshold adopts a fixed threshold, which is prone to misclassification in some areas.
  2. It is necessary to build the grid many times, which may reduce some work efficiency
  3. For some scenes, the filtering effect may not be very good.

Its advantages are explained in the 550th line of the 24th page. Compared with other methods, this method can also achieve a good filtering effect for the multi-layer building scene which is difficult to deal with by grid filtering. Compared with multiple sample data, this method has better stability and accuracy than other methods.

Point 3: In the conclusion section, the limitations of the proposed method must be discussed by the authors. 

Response 3: Thanks for your suggestion, on page 24, line 559, We have described the limitations of the current method, and on this basis, the next research work is determined.

Point 4:  In the conclusion section, the authors will need to clearly address their research contributions in theory.

Response 4: Thanks for your suggestion, on page 24, line 550, we briefly explain the research contribution of this method, and verify that the clustering method can be applied to the scene point cloud filtering and can achieve good results. Through multiple sets of experimental comparisons, the stability and accuracy of this method in each scene are determined. The theoretical support of this method is also elaborated in the discussion section.

This manuscript is a resubmission of an earlier submission. The following is a list of the peer review reports and author responses from that submission.

Round 1

Reviewer 1 Report

The paper charts are not standardized. The description of innovation points in the paper is not clear.

Reviewer 2 Report

In the paper, authors proposed a Triangular Grid Filter based on Slope Filter to separating ground points from non-ground points. There are four steps about the method: First, authors give several types of error. First, the slope filter was used to remove non-ground points to reduce the second type of error. Second, the triangular grids are constructed to determine violation-triangule through the grid. Third, the three-point colliner method was used to extract the regular points in the violation points. Finally, the clustering and convex hull algorithm was used to remove the disperse points. The experiments show that the the proposed method has a high processing efficiency and accuracy.   The paper is interesting. However, a few things are to be considered sincerely before its publication.

1. In the abstract section, reduce Ⅱ error, what is Ⅱ error.

2. In the last step, authors use clustering algorithm to remove disperse points and irregular landmarks. This is an important step. However, there are no introduction about the latest clustering algorithm in the section of introduce, for example

Chen X, Wu H, Lichti D, et al. Extraction of indoor objects based on the exponential function density clustering model[J]. Information Sciences, 2022, 607: 1111-1135. 

A. Rodriguez and A. Laio. Clustering by fast search and find of density peaks, Science, 344(6191) (2014) 1492-1496. 

3. The language should be improved, such as:

Line 166: the principle of it is that if the distance between points is less than the distance

Line 170: KD-Tree uses the method of what method?

Line 177: randomly select a point from the set of points should be passive voice

Line 320: From this table”, which table?

Line 455:”Compared with using the grid instead of triangular grid,triangular grid can more”

4. Line 233: The clustering results are distinguished by color after grouping(Figure 12).  Is it Figure 12 or Figure 11.

How do you obtain the color. If the device directly the color, it is difficulty to obtain the clustering results, as showing in Figure 11. What cluster method is used to conduct the clusteing. The point cloud with color information should be given with a figure.

5. The Euclidean clustering algorithm involves the KD-Tree method. What does this meaning? What is the Euclidean clustering algorithm, it is k-means clusteing, DBSCAN clustering or density peak clustering? Why use this clustering algorithm. Why not use the latest clustering algorithm.

Reviewer 3 Report

In this paper a method which classifies 3D points into ground and non-ground points, is presented.

The paper is impossible to be read due to poor English language. The authors should rewrite the paper.

However, here are some thoughts.

First and foremost, the references are strongly outdated i.e., from the 21 available references the 14 are before 2013 with the most of them around 2000.

The steps included in the proposed algorithm are:

·       Firstly, a triangular grid of the area is constructed.

·       Then the algorithm detects the adjacent triangles in the grid and calculates the normals

·       If the angle between the normals is larger than a predefined threshold value, the algorithm marks the adjacent triangles as violation triangles.

·       The longest edge of the violation triangles is extracted. If the longest edge exceeds a predefined threshold value, the lowest point of the triangle is marked as regular violation point.

·       Then, using the three-point collinearity equation the algorithm determines whether the regular violation points are collinear or not. The collinear points are marked.

·       The regular violation points are used as centers of the clustering method. Two additional threshold values (clustering threshold and elevation threshold) are included into this step.

·       Several steps inside the clustering and in general are repeated using a different set of threshold values each time.

The proposed approach is too complicated and it uses many threshold values which are selected manually and so I believe that the proposed method is not efficient. The code implementation is not open and so I am not able to validate the efficiency of the code. No information is provided for the manual threshold values selection process.

Among others, Section 1 should be renamed from Introduce to Introduction. There are very large, hard to read sentences e.g., “Thirdly,according to the three-point collinear method to 20 extract the regular points in the violation points and the regular non-ground points are remved by 21 these regular points”. Some areas in the paper maybe are probably copy-pasted, so many confusing words like “remvingremved” instead of removing and removed dominate the article. Additionally, space should be inserted after commas, full stops etc.

To sum up I believe that the article requires many changes which cannot be made within a reasonable time framework.
